# A Numerical Study of the Wind Speed Effect on the Flow and Acoustic Characteristics of the Minor Cavity Structures in a Two-Wheel Landing Gear

**Longlong Huang** [1], **Kun Zhao** [1,*], **Junbiao Liang** [1], **Victor Kopiev** [2], **Ivan Belyaev** [2] and **Tian Zhang** [3]

[1]   Key Laboratory of Aerodynamic Noise Control, China Aerodynamics Research and Development Center (CARDC), No. 6, Sounth Section, Second Ring Road, Mianyang 621000, China; hlong9925@163.com (L.H.); lsc_air@126.com (J.L.)

[2]   Acoustic Division, Central Aerohydrodynamics Institute (TsAGI), Radio Str. 17, 105005 Moscow, Russia; vkopiev@mktsagi.ru (V.K.); aeroacoustics@tsagi.ru (I.B.)

[3]   China Academy of Engineering Physics, No. 64, Mianshan Road, Mianyang 621000, China; Wch410204603@126.com

*   Correspondence: zhaokun@cardc.cn

**Abstract:** The landing gear is widely concerned as the main noise source of airframe noise. The flow characteristics and aerodynamic noise characteristics of the landing gear were numerically simulated based on Large Eddy Simulation and Linearized Euler Equation, and the feasibility of the simulation model was verified by experiments. Then the wind speed effect on the flow and acoustic characteristics of the minor cavity structures in a two-wheel landing gear were analyzed. The results show that the interaction of vortices increases with the increase of velocity at the brake disc, resulting in a slight increase in the amplitude of pressure fluctuation at 55 m·s$^{-1}$~75 m·s$^{-1}$. With the increase of speed, the obstruction at the lower hole of torque link decreases, and many vortical structures flow out of the lower hole and are dissipated, so that the pressure fluctuation amplitude of 75 m·s$^{-1}$ almost does not increase relative to 55 m·s$^{-1}$. The contribution of each part in the landing gear to the overall noise is as follows: shock strut > tire > torque link > brake disc. At the speed of 34 m·s$^{-1}$~55 m·s$^{-1}$, the contribution of each component to the total noise increases with the increase of speed, and the small components such as torque link and brake disc contribute more to the total noise. At the speed of 55 m·s$^{-1}$~75 m·s$^{-1}$, the increase of overall noise mainly comes from the main components such as shock strut and tire, and the brake disc and torque link contribute very little to the overall noise. It provides a reference for the further noise reduction optimization design of the landing gear.

**Keywords:** landing gear; aerodynamic noise; large eddy simulation; linear euler equation

## 1. Introduction

With the progress of aircraft engine noise reduction technology, airframe noise has been identified as a new challenge to the aeronautical industry [1]. The main source of the airframe is the landing gear [2], followed by the high lift devices. Therefore, the landing gear has been widely studied [1–5].

In the early stage, in order to identify the most dominant sources of the landing gear, the European research program RAIN carried out full-scale aeroacoustics tests in the German Dutch wind tunnel (DNW) on the landing gear of A340 aircraft [3,6,7]. Based on experimental research, add-on noise reduction technology was developed to realize the overall noise reduction up to 3 dB [8]. However, this noise reduction method cannot be applied to current aircraft in the short term. In order to take into account the noise impact in the design stage, the European research project SILENCER ("Significantly Lower Community Exposure to Aircraft Noise") was committed to how to design low-noise landing gear in the development stage [4]. Then a European Co-financed project called TIMPAN ("Technology to IMprove Airframe Noise") was launched [2]. It mainly studied some more

advanced low-noise designs for aircraft landing gear and the high lift wing. At present, there are still many projects to study airframe noise [9–12], fairing [13] technology has been widely studied, and some novel noise reduction approach such as air curtain [14,15] and wire mesh screen model [16] have been proposed. For further information, please refer to a review paper by Zhao et al. [17].

As a fundamental scenario aeroacoustics simulation, tandem cylinders have been studied by many researchers. For instance, different turbulence models were utilized to conduct to numerical simulate [18,19], and the results were compared with the experimental data of NASA achieved from Basic Aerodynamic Research Tunnel (BART) and Quiet Flow Facility (QFF) [20–22]. In addition, Dawi [23] used numerical approach to study the acoustic performance of two square cylinders.

A large number of landing gear experiments provide a comprehensive data set for numerical simulation [7]. At the beginning, the flow field simulation of landing gear mainly adopts RANS (Reynolds-Averaged Navier-Stokes) method [5]. The hybrid RANS/CAA approach was used for the simulation of broadband sound generation [24]. Then, owing to the DES [25–27] (Detached Eddy Simulation) and LES (Large Eddy Simulation) [28–30] ability to capture the small-scale turbulence characteristics related to high-frequency noise, they have been widely used to solve the landing gear flow field. Many researchers have developed different new methods on the basis of LES and DES [31–34]. Ribeiro [35] used the LBM (Lattice Boltzmann Method) and FW-H (Ffowcs Williams-Hawkings) methods to study the effect of the grid on the aeroacoustics characteristics of the landing gear. Francois Fortin [26] compared CFD simulation with experiments in detail to verify the flow field, thereby evaluating the accuracy of CFD results for noise prediction. At present, the main noise radiation simulation methods are FW-H, APE (Acoustic Pertubation Equations) [36], and LEE (Linearized Euler Equation) [37,38]. The Linear Euler method has been used to solve the sound propagation problems in the near and intermediate sound fields [39]. Some researchers have derived the convection form of the source term and verified the correctness of the source term representation method [40], Bissuel [37] used linear N-S equations to calculate the noise problem of jet engine blades. Christophe Bailly [38] used linear Euler equations to study the numerical solutions of monopole, dipole, and quadrupole noise and verified them through analytical solutions. At present, there is relatively little research on the application of LES/LEE to landing gear.

Analyzing the flow characteristics of landing gear is the key to solving the problem of aerodynamic noise of landing gear [3]. Some researchers have conducted a detailed study on the development and evolution of the landing gear trailing vortex [41]. Due to the complex structure of the landing gear, some researchers have carried out research on the noise radiation of various parts of the landing gear, such as tires, struts, and torque linkages, etc. [42]. Many studies found that the tonal noise of landing gear mainly originates from flow induced oscillations in cavities [5], so the flow and noise of the cavity structure of the landing gear, such as the hub cavity and the rim cavity, have been widely concerned [43]. However, there are relatively few numerical studies of the wind speed effect on the flow and acoustic characteristics of the minor cavity structures in two-wheel landing gears.

Firstly, the landing gear test was carried out based on FL-17 wind tunnel (Section 2), and then the landing gear numerical model was established based on LES/LEE method, which is verified by experiments (Section 3). Then the effects of wind speed on the flow and acoustic characteristics of the minor cavity structures in two-wheel landing gears (brake disc and torque link) are further analyzed (Section 4).

## 2. Setup of Landing Gear Experiment

### 2.1. FL-17 5.5 m × 4 m Aeroacoustic Wind Tunnel

The test was carried out in the 3/4 opening test section of the 5.5 m × 4 m aeroacoustic wind tunnel at the China Aerodynamics Research and Development Center (CARDC), as shown in Figure 1. The test section is surrounded by a 26 m × 18 m × 27 m anechoic chamber, and anechoic wedge is installed around the anechoic chamber to suppress sound

reflection. The air flows out from the nozzle of 5.5 m (width) × 4 m (height), and the incoming flow velocity ranges from 0~100 m·s$^{-1}$. When the wind speed of the opening section is 80 m·s$^{-1}$, the turbulent flow energy at the center of the nozzle is less than 0.05%. In the test, the landing gear was installed on a test platform measuring 13.5 m (length in the direction of incoming flow) × 8.6 m (width) × 6 m (height).

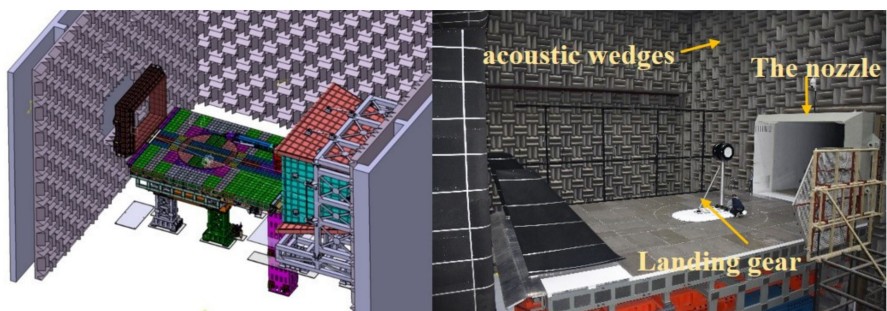

**Figure 1.** Schematic diagram of 5.5 m × 4 m aeroacoustic wind tunnel.

### 2.2. Landing Gear Model

The model of the landing gear is shown in Figure 2, which mainly includes pillars, tires, struts, and other components. *O* is the coordinate origin (the midpoint of the connecting line between the two tire centers of the landing gear). In the test, the entire model was connected to the base through the connecting shaft at the bottom of the outer cylinder, and the base is fixedly connected to the test support platform. In order to avoid additional noise, both the base and the connecting shaft were equipped with fairings to achieve a smooth transition.

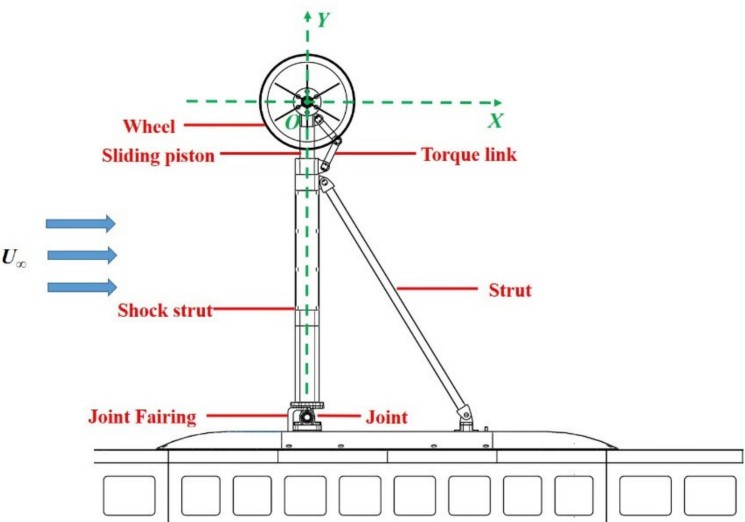

**Figure 2.** Landing gear model.

### 2.3. Acoustic Measurement Arrangement

The acoustic test system to obtain the aerodynamic noise characteristics of the landing gear, consists of 21 free-field microphones and 135 microphone arrays. Figure 3 shows the schematic diagram of the microphone installation. The parameters of the types of instruments and equipment required during the test are shown in Table 1.

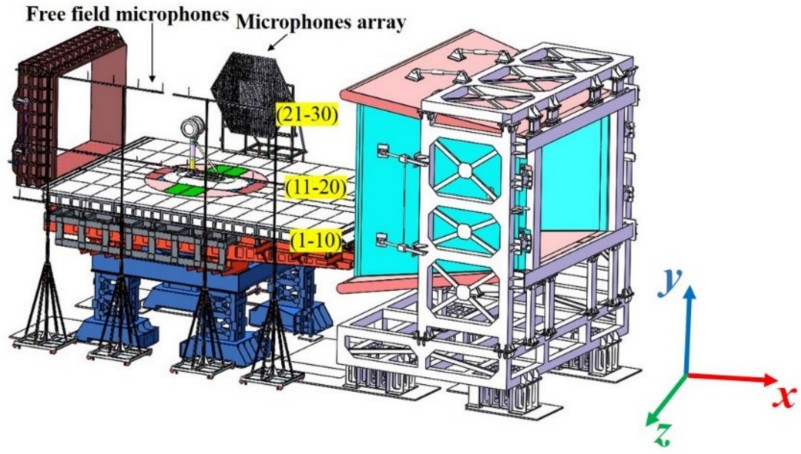

**Figure 3.** Microphone distribution map.

**Table 1.** Experimental equipment parameters.

| Experimental Equipment | Device Model | Performance Parameter |
|---|---|---|
| Acoustic wind tunnel | 5.5 m × 4 m | (1) Driven by an impeller with a power of 12.5 MW; (2) Wind speed range: 0~100 m·s⁻¹; (3) The turbulence in the center of the opening section of the wind tunnel nozzle is 0.05%; (4) The background noise is 75.6 dB (A) |
| Far-field microphone | G.R.A.S 46AE | (1) Frequency range: 3.15 Hz~20 kHz; (2) Sound pressure dynamic response: 14 dBA re.20 μPa~135 dB re.20 μPa; (3) Sensitivity: 50 mV/Pa, 250 Hz |
| Array microphone | 135 G.R.A.S 40PH | (1) Frequency range: 100 Hz~20 kHz; (2) Dynamic response of sound pressure: 32~135 dB (A); (3) Sensitivity: 50 mV/Pa, 250 Hz |
| data collection systems | NI PXle-4499 | Sampling frequency 51.2 kHz |

## 3. The Setup of the Simulation

### 3.1. Computational Domain

The landing gear is located 8.3 $D_L$ away from the wind tunnel nozzle, the incoming wind speed $U_\infty$ = 34~75 m·s⁻¹, the Mach number $M$ is 0.099~0.219, $D_L$ = 720 mm is the diameter of the landing gear tire. When the velocity is 34 m·s⁻¹~75 m·s⁻¹, the corresponding Reynolds number Re = $\rho U_\infty D_L / \mu$ is $1.7 \times 10^6$~$3.7 \times 10^6$. The calculation domain is 27.7 $D_L$ (length) × 18.1 $D_L$ (width) × 8.3 $D_L$ (height), as shown in Figure 4, $O$ is the coordinate origin (the midpoint of the connecting line between the two tire centers of the landing gear), and $U$, $V$, and $W$ are the velocity components in $X$, $Y$, and $Z$ directions respectively.

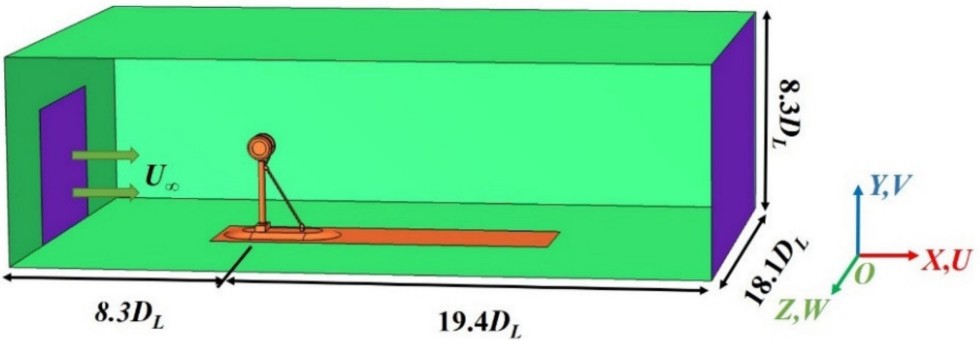

**Figure 4.** Flow domain.

### 3.2. Governing Equation

CFD is mainly based on Smagorinsky model in LES module of COMSOL platform. The LES equation [28] of isothermal incompressible flow is:

$$\frac{\partial \overline{u}_i}{\partial x_i} = 0 \tag{1}$$

$$\frac{\partial \overline{u}_i}{\partial t} + \frac{\partial \overline{u}_i \overline{u}_j}{\partial x_j} = -\frac{1}{\rho}\frac{\partial \overline{p}}{\partial x_i} + v\frac{\partial^2 \overline{u}_i}{\partial x_j x_j} - \tau_{ij}\frac{\partial \tau_{ij}}{\partial x_j} \tag{2}$$

where $u_i$ is the velocity component and $\overline{u}_i$ is the filtered average velocity component, $\overline{p}$ is the filtered average pressure of the fluid, $\rho$ is the density of the fluid, $v$ is the viscosity.

The strain tensor $\overline{S}_{ij}$ is expressed as follows

$$\overline{S}_{ij} = \frac{1}{2}\left(\frac{\partial \overline{u}_i}{\partial x_j} + \frac{\partial \overline{u}_j}{\partial x_i}\right) \tag{3}$$

The compressive lattice stress is expressed as

$$\tau_{ij} = \overline{u_i u_j} - \overline{u}_i \overline{u}_j \tag{4}$$

The approximate formula of Boussinesq is

$$\tau_{ij} - \frac{\delta_{ij}}{3}\tau_{kk} = -2v_{sgs}\overline{S}_{ij} \tag{5}$$

where $v_{sgs}$ is the eddy viscosity model, the formula is

$$v_{sgs} = C\Delta^2 |\overline{S}|\overline{S}_{ij} \ \ |\overline{S}| = \left(2\overline{S}_{ij}\overline{S}_{ij}\right)^{1/2} \tag{6}$$

The sound propagation of landing gear was studied based on LEE method [38,40] of COMSOL platform. The governing equation is as follows:

$$\frac{\partial \rho_t}{\partial t} + \nabla\cdot(\rho_t \mathbf{u}_0 + \rho_0 \mathbf{u_t}) = \mathbf{S_c} \tag{7}$$

$$\frac{\partial \mathbf{u_t}}{\partial t} + \left(\left(\mathbf{u_t} + \frac{\rho_t}{\rho_0}\mathbf{u_0}\right)\cdot\nabla\right)\mathbf{u}_0 + (\mathbf{u_0}\cdot\nabla)\mathbf{u_t} + \frac{1}{\rho_0}\nabla p_t = \mathbf{S_m} \tag{8}$$

where $\rho_t$, $\mathbf{u_t}$, and $p_t$ are the small perturbations exerted by density, velocity, and pressure on the mean flow, respectively; $\rho_0$, $\mathbf{u_0}$, $p_0$ are the density, velocity, and pressure of the flow field respectively.

The $\mathbf{S_c}$ and $\mathbf{S_m}$ are the mass source term and momentum source term, respectively.

### 3.3. Meshing

Figure 5a,b show the three-dimensional grid distribution of the landing gear and the grid division of *YZ* cross-sections, respectively. The meshes around the landing gear and the wake area, whose minimum mesh size was 2.5 mm, were refined to capture the wake vortex formed by the circumfluence, and it was appropriately increased in the area far away from the landing gear. A prismatic layer mesh was used for the wall of the landing gear, which the $Y^+ < 10$, 14, and 18 correspond to wind speeds of 34 m·s$^{-1}$, 55 m·s$^{-1}$ and 75 m·s$^{-1}$ respectively, a total of 22 layers are divided, and the mesh growth rate in the direction of the boundary layer thickness was 1.18.

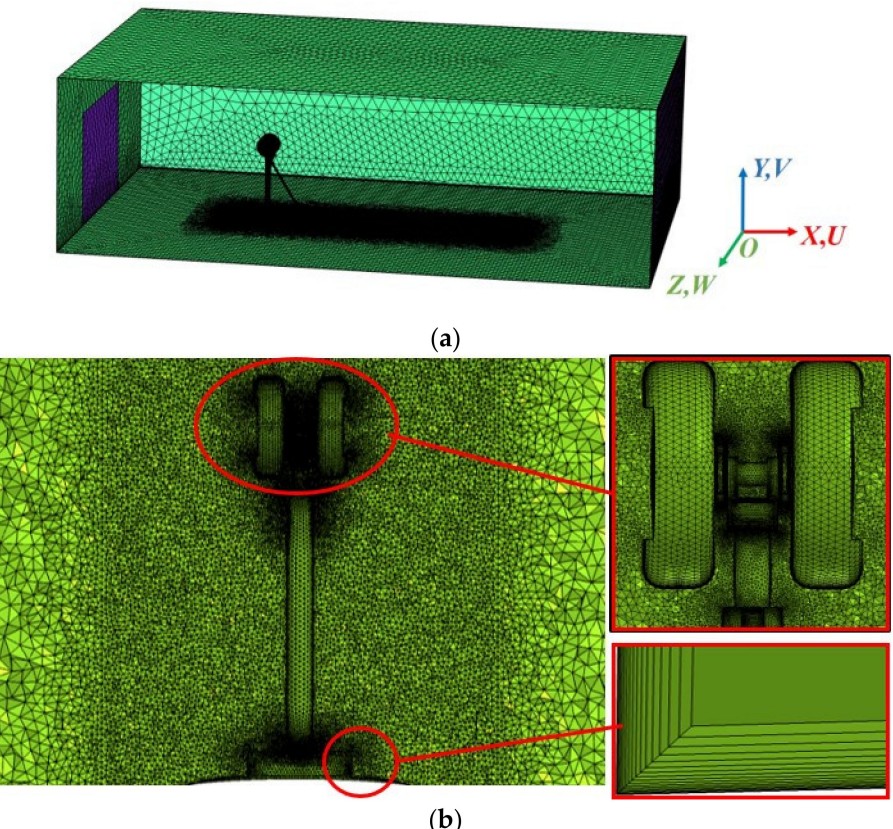

**Figure 5.** Flow field grid of landing gear. (**a**) three-dimensional grid (**b**) grid division of *YZ* cross-sections.

### 3.4. Simulations Parameters

The inlet adopted the velocity condition ($U = U_\infty$, $V = W = 0$), the outlet adopted the open boundary condition (normal stress $f_0 = 0$), the left and right sides adopted the periodic boundary condition ($\mathbf{u}_{src} = \mathbf{u}_{dst}$), the bottom surface and the landing gear adopted the non-slip wall ($\mathbf{u} = \mathbf{0}$), and the top adopted the slip wall surface condition ($\mathbf{un} = \mathbf{0}$). For acoustic calculations, the perfect matching layer was used to simulate the non-reflective boundary of the model except for the bottom surface, and the hard sound field boundary was used to simulate wall reflection on the surface and bottom of the landing gear [37].

Based on the coarse grid calculation of 0.3 s, the result of the coarse grid was taken as the initial value of the fine grid for 0.1 s. The time step of analyzing the transient flow was $2.5 \times 10^{-5}$ s, which is determined by the formula $\Delta t = C_{FL}\Delta x/U_\infty$, $\Delta x = 0.0025$ m is the size of minimum mesh, $C_{FL}$ set to 0.75. The maximum frequency of the sound field was 2000 Hz and the frequency interval was 10 Hz.

### 3.5. Verification of Calculation Method

Many structures of landing gear can be compared with cylinders. As a fundamental calculation example of aeroacoustics, the tandem cylinder numerical model was built and the results were compared with the experiments. As shown in Figure 6, it is the model diagram of tandem cylinders, the diameter *D* of the cylinder is 0.05715 m, the distance between the centers of the tandem cylinders is 3.7 *D*, the spanwise (the *z* direction) length is 4 *D*, and the incoming flow velocity $U_\infty$ is 44 m·s$^{-1}$.

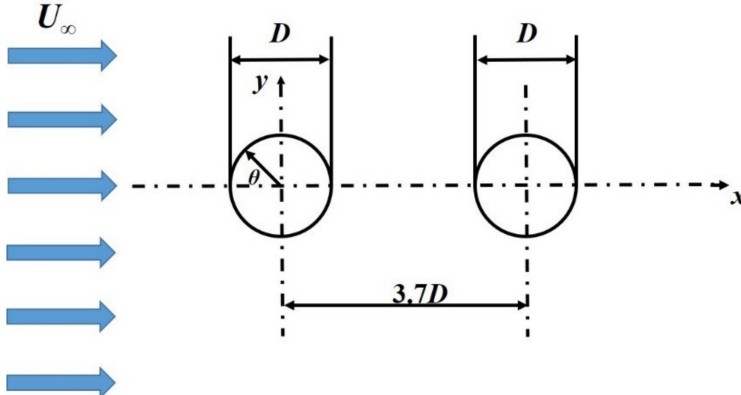

**Figure 6.** Schematic diagram of flow around tandem cylinders.

The inlet adopted velocity inlet ($U = U_\infty$, $V = W = 0$), the outlet was set as atmospheric pressure ($p = 1$ atm), the cylindrical surface was non slip wall ($\mathbf{u} = \mathbf{0}$), the boundaries perpendicular to the $z$ direction were set as asymmetric boundary ($\mathbf{u}_{src} = \mathbf{u}_{dst}$), and other boundaries were set as slip wall ($\mathbf{un} = 0$). In the calculation domain, the grids around the cylinder and the wake area were encrypted, and a total of 3.21 million grid elements were divided. The time step was $5 \times 10^{-5}$ s, and a total of 0.2 s was calculated. In Figure 7, the boundary condition setting and meshing of tandem cylinder model are shown.

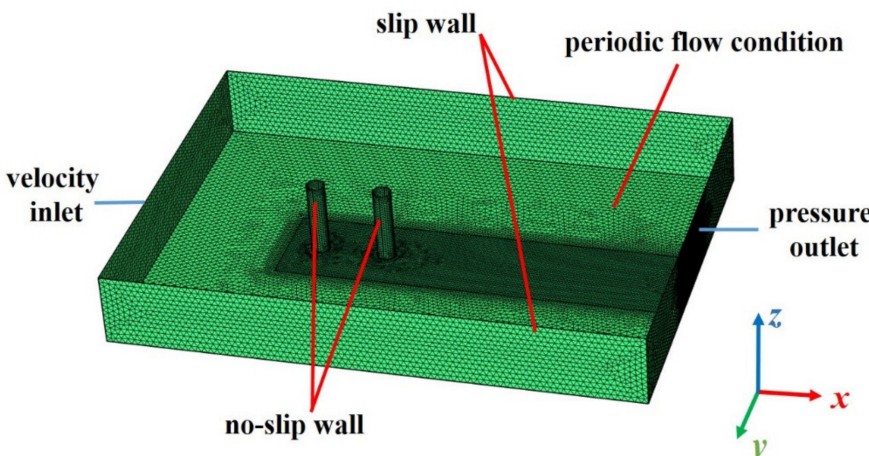

**Figure 7.** Grid and boundary conditions.

### 3.5.1. Surface Pressure Coefficient

Figure 8 shows the numerical results of time average pressure coefficient distribution on the surface of upstream cylinder and downstream cylinder, which are compared with the experimental results of QFF and BART [20–22]. In the figure $\theta$ is the azimuth.

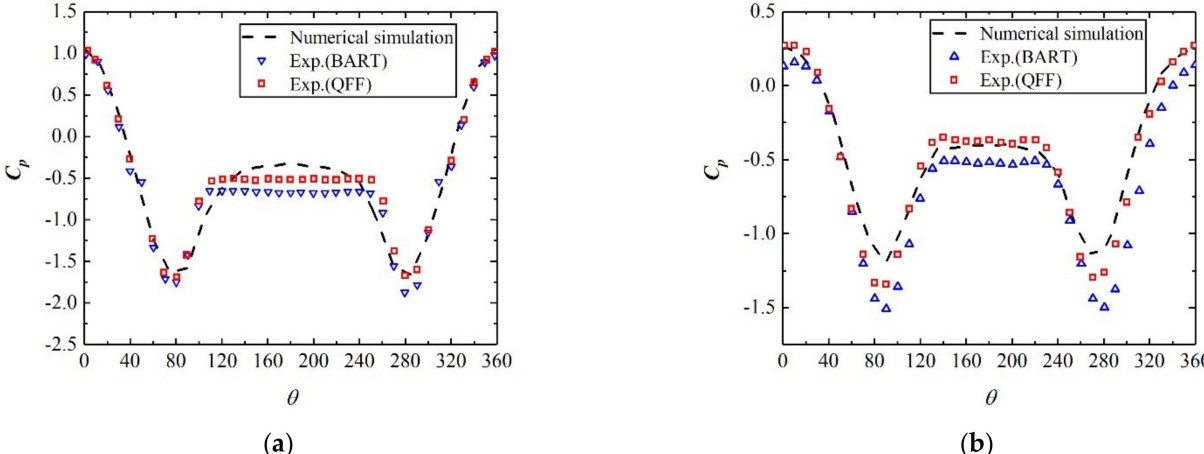

**Figure 8.** Comparison of pressure coefficients. (**a**) Upstream cylinder (**b**) Downstream cylinder.

It can be seen from Figure 8 that the surface pressure coefficient of the upstream cylinder is in good agreement with the QFF and BART test results, and the downstream cylinder surface pressure coefficient is closer to the QFF experimental results. This is consistent with the results of literature [18,19]. The main reason for the error at position 80° and 280° of the downstream cylinder is that the downstream cylinder is affected by the upstream cylinder, so the flow is more complex, and the spanwise length in the simulation is inconsistent with the experiment.

3.5.2. Far Field Noise

Some researchers have tested the aerodynamic noise of flow around tandem cylinders based on QFF wind tunnel, and three far-field microphones were arranged in the experiment. The coordinates of the three measuring points are A (9.11 *D*, 32.49 *D*), B (–8.33 *D*, 27.815 *D*), and C (26.55 *D*, 27.815 *D*). The three points are located on the central section of the cylinder. Figure 9 shows the comparison of the PSD (Power Spectral Density) of evolution with $S_t$ between experiment and simulation at points A, B, and C.

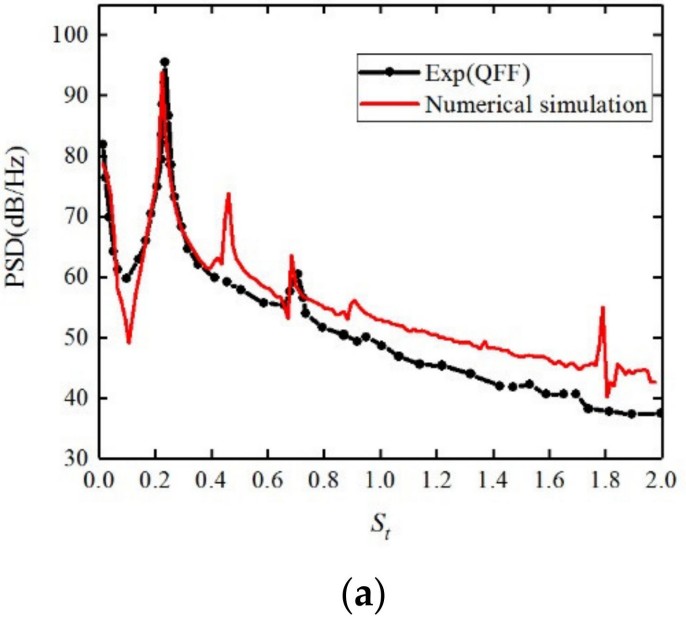

(**a**)

**Figure 9.** *Cont.*

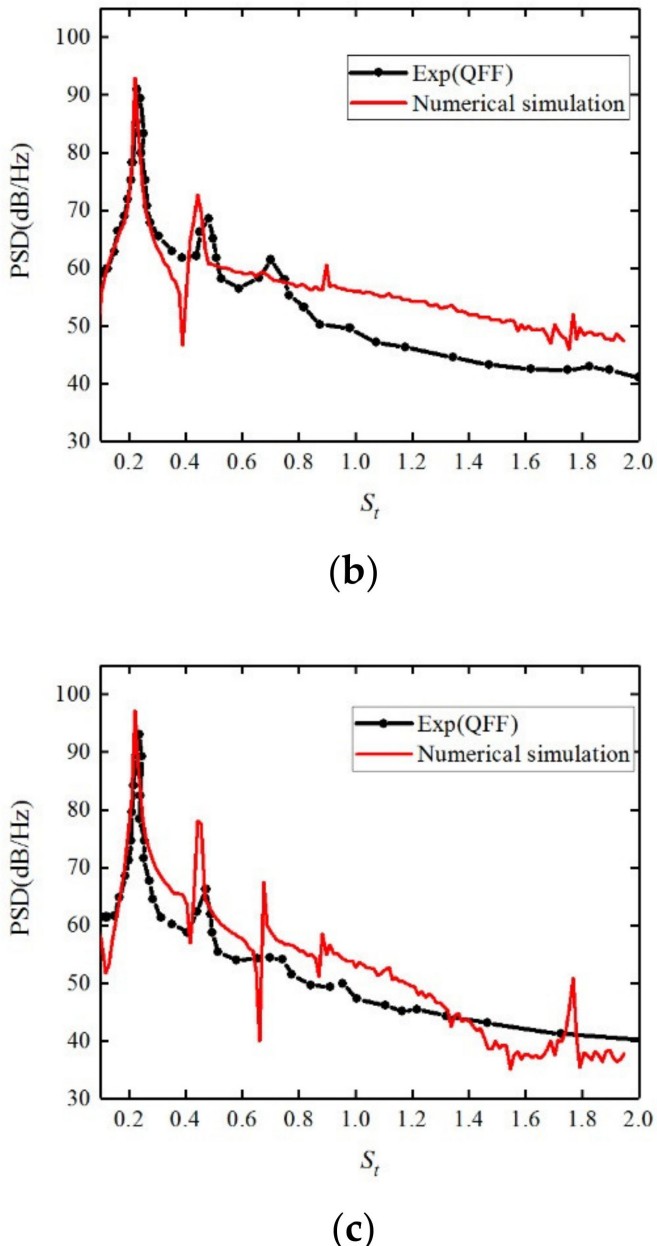

**Figure 9.** Comparison of PSD evolution with $S_t$ between experiment and simulation. (**a**) point A (**b**) point B (**c**) point C.

Through the analysis of the Figure 9, it is concluded that the simulation results at the three observation points are in good agreement with the experimental results, and the peak frequency and amplitude can be accurately captured. The Strouhal numbers corresponding to the maximum peaks of points A, B, and C measured by test and simulation are $S_t = 0.234, 0.227, 0.240$ and $S_t = 0.224, 0.221, 0.221$, respectively. The simulation results at the highest peak are in good agreement with the experimental results, but there are some differences in other positions. Furthermore, from Figure 8, it can be found that the downstream cylinder is affected by the upstream cylinder, and the numerical model is scaled in the spanwise direction relative to the experiment. This makes the pressure coefficient of the downstream cylinder different from the experimental results, resulting in differences between the simulation and experimental values in the far-field sound pressure level. Compared with the results of flow and acoustics, the numerical model is feasible.

## 4. Results and Discussion

### 4.1. Verification of Landing Gear Calculation Model

Figure 10 shows the iso-surface map of the vortex formed by the flow around the landing gear based on the $Q$ criterion when the incoming flow velocity is 75 m·s$^{-1}$. The color indicates the change of Mach number. It can be seen from the figure that complex vortical structure is formed at the tail of the landing gear, thereby forming a complex flow field change, which generates aerodynamic noise.

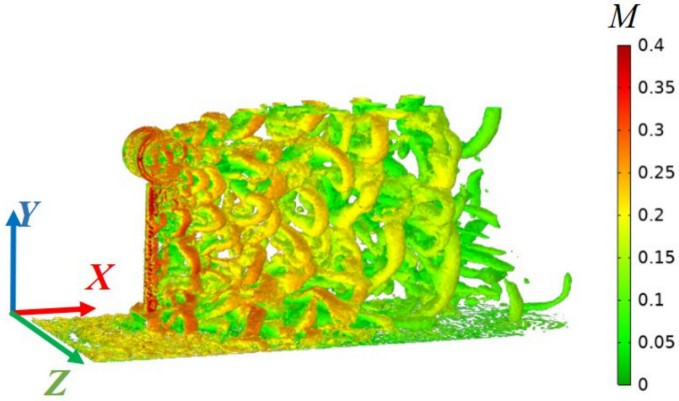

**Figure 10.** $Q$ iso-surface map ($Q = 3000$ s$^{-1}$).

Figure 11 shows the comparison of the SPL (Sound Pressure Level) evolution with the frequency between test and numerical simulation under different wind speeds. Figure 11a–c are the result of microphone 13, and Figure 11d–f are the result of microphone 15. Table 2 is the comparison the OASPL (Overall Sound Pressure Level) between test and simulation.

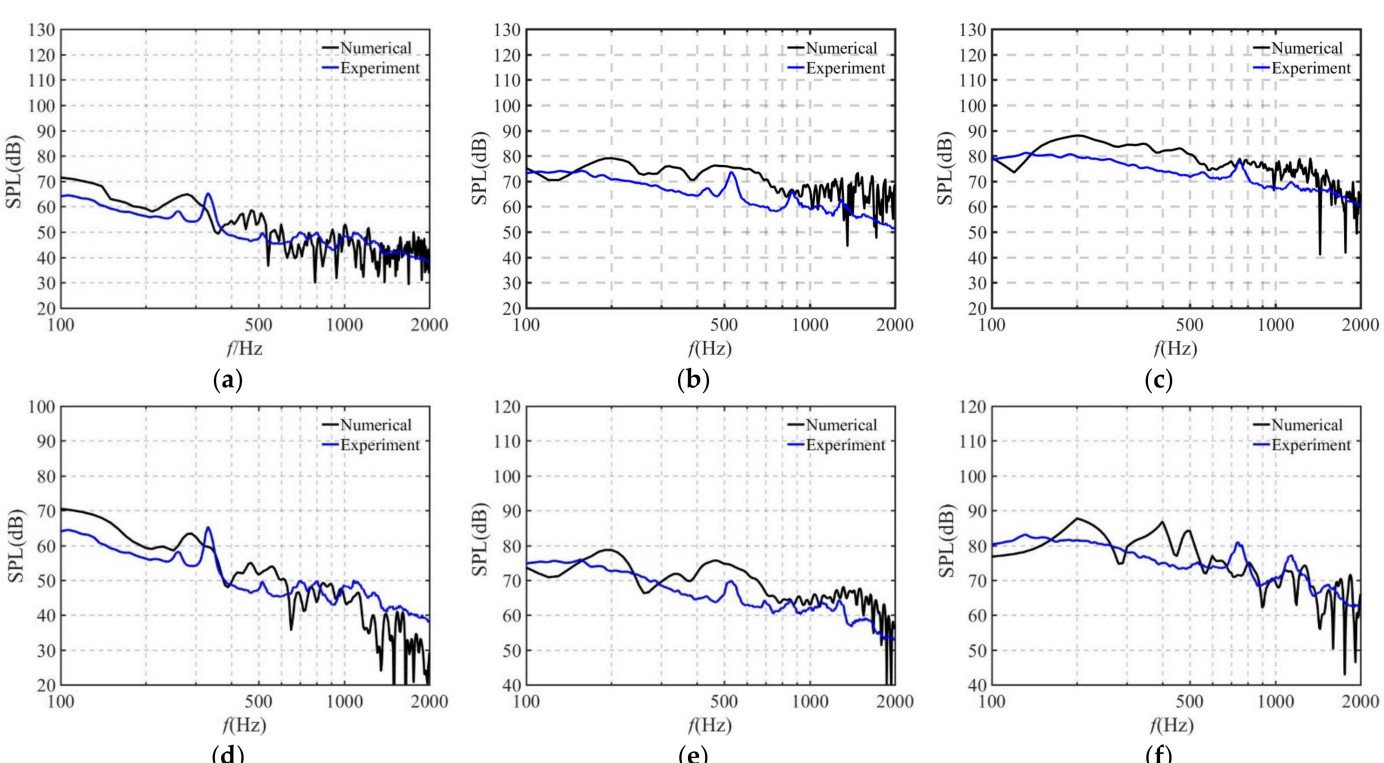

**Figure 11.** Comparison of SPL evolution with the frequency between test and simulation. (**a**) 34 m·s$^{-1}$ (**b**) 55 m·s$^{-1}$ (**c**) 75 m·s$^{-1}$ (**d**) 34 m·s$^{-1}$ (**e**) 55 m·s$^{-1}$ (**f**) 75 m·s$^{-1}$.

**Table 2.** Comparison of OASPL between test and simulation.

| | Velocity | Experiment (15#/13#) | Simulation (15#/13#) | Difference/% (15#/13#) |
|---|---|---|---|---|
| OASPL/dB | 34 m·s$^{-1}$ | 78.7/75.8 | 79.1/79.06 | 0.5%/4.3% |
| | 55 m·s$^{-1}$ | 92.5/88.7 | 92.9/92.6 | 0.4%/4.4% |
| | 75 m·s$^{-1}$ | 101.4/96.9 | 100.1/99.4 | 1.2%/2.6% |

From Figure 11, it can be concluded that the difference of spectrum amplitude between the experimental and simulation are minor, but there are some differences in the wave peaks corresponding to the frequency, mainly due to the complex flow of the landing gear and the fact that the model is not scaled, so more grids are needed to accurately analyze the flow field. Moreover, LES can capture small flow structures, which makes it difficult to extract sound source, resulting in multiple wave peaks in the numerical results. It can be seen from Table 2 that the total far-field sound pressure level difference of landing gear at different wind speeds is within 5%, which meets the research on the amplitude characteristics of landing gear aerodynamic noise at different wind speeds.

Figure 12 shows the comparison of sound pressure levels under different wind speeds. According to Figure 12, it can be concluded that with the increase of wind speed, the fluctuation frequency and amplitude of sound pressure increase, but the amplitude increment of 34 m·s$^{-1}$~55 m·s$^{-1}$ is greater than that of 55 m·s$^{-1}$~75 m·s$^{-1}$.

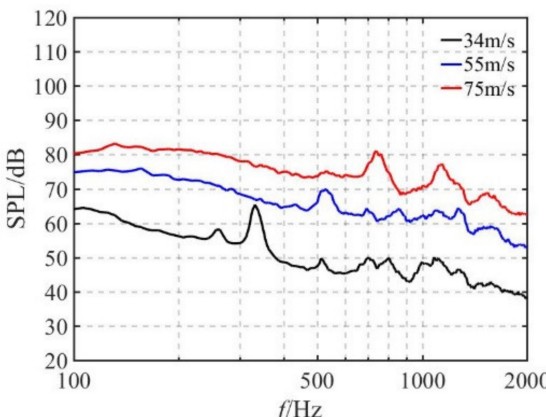

**Figure 12.** Comparison of SPL under different wind speeds.

In order to further analyze the relationship between the velocity and the acoustics, scaling was used to predict the effects on acoustic performance from other factors in addition to $U_\infty$. The Scaled SPL is as follows:

$$SPL_{scaled} = SPL - 10 \times n \times \log(U_\infty/U_{ref}) \tag{9}$$

where $U_{ref}$ = 65 m·s$^{-1}$ is the reference velocity of incoming flow, $n$ is the power index of the scaling proportionality. It is generally known that the landing gear is a dipole-like source due to unsteady pressure forces so $n$ = 6 is preferred. However, it is inferred that $U_\infty^7$ only works for a compact model and a higher order of scaling is required due to the existence of smaller geometric details in a real landing gear [9]. Therefore, $n$ = 6, 7 were utilized.

The scaled spectra of experiment using proportionality of $U_\infty^6$ and $U_\infty^7$ is shown in Figure 13a,b. It is observed that scaling proportionality of $U_\infty^7$ shows better performance for the frequency over 150 Hz. For the low frequency range below 150 Hz, $U_\infty^6$ appears to be slightly better. In contrast, Figure 13c,d are the scaled spectra of simulation. It can be conducted that the $U_\infty^7$ performs better than $U_\infty^6$. In conclusion, the model here is much larger, and with minor details. Therefore, the seven-power law is more appropriate for the scaling of large-size, while the six-power law is appropriate for the small-size.

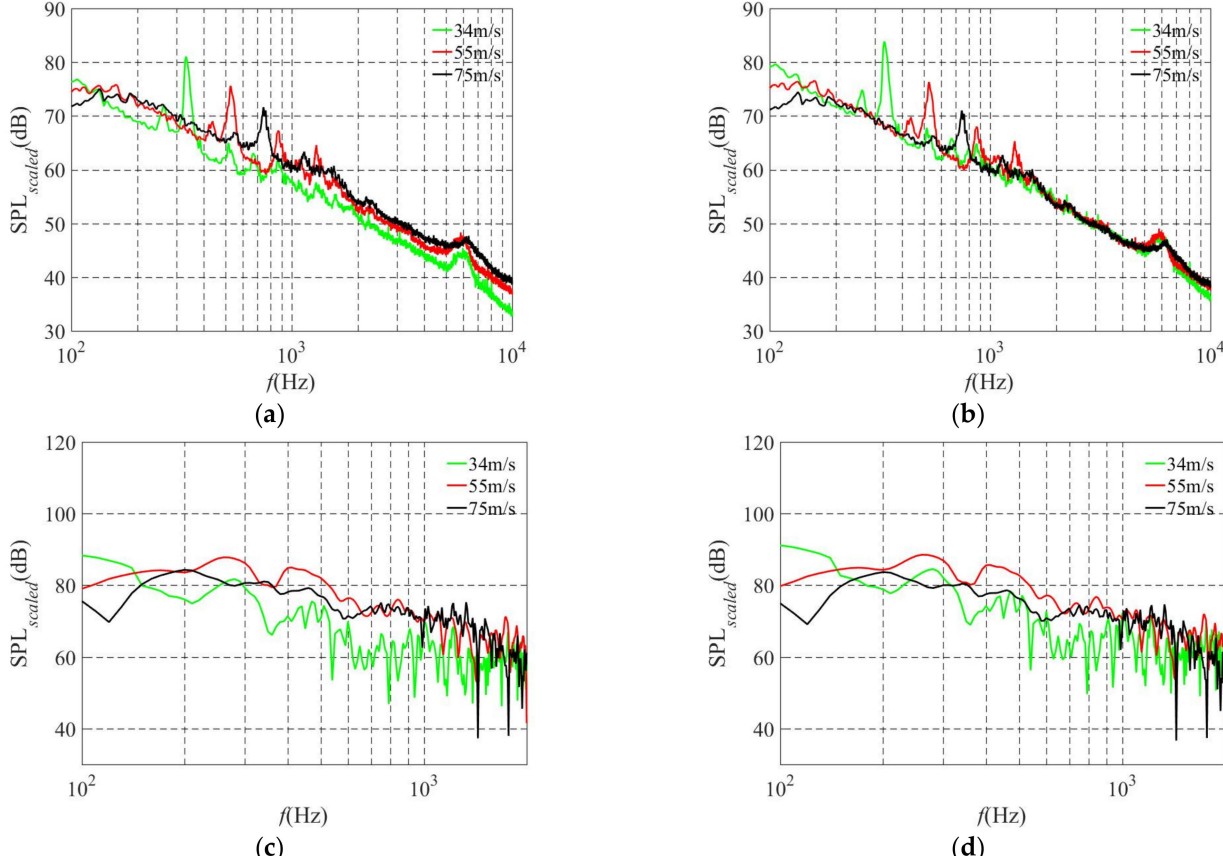

**Figure 13.** Scaled spectra of experiment and simulation at MIC.13. (**a**) $n = 6$ (**b**) $n = 7$ (**c**) $n = 6$ (**d**) $n = 7$.

Due to the complex structure of the landing gear, it is difficult to analyze the flow of the whole landing gear. In addition, flow of small parts of the landing gear is more complex and has a great impact on the overall noise of the landing gear. Furthermore, the wind speed effects on the flow and acoustics of minor cavity structures in a two-wheel landing gear were analyzed.

### 4.2. Flow Characteristics of Brake System and Torque Link

Figure 14 is structural diagram of the cavity of the brake system and torque link of the landing gear. Figures 15 and 16 respectively show the streamline and vorticity magnitude distribution of brake disc and torque link. The flow at these two parts is relatively complex, and many vortical structures are formed inside. In order to analyze the internal flow changes, some points were selected at the center and tail of brake disc and named them $B_a$ and $B_b$ respectively, and other points were selected on the upper hole, internal center and lower hole of the torque link and named them $T_a$, $T_b$, and $T_c$ respectively.

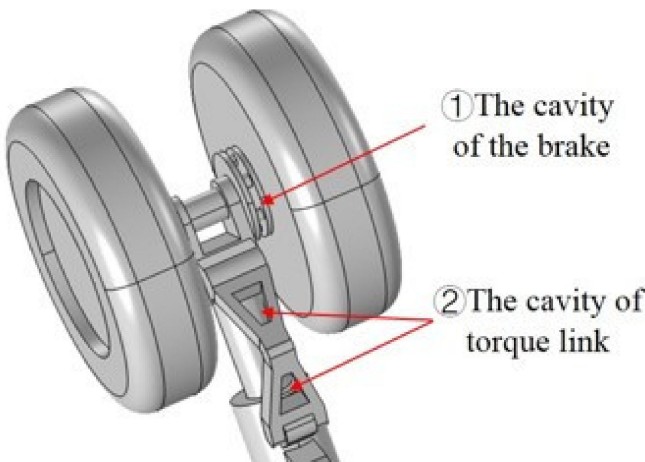

**Figure 14.** Brake system and torque link cavity structure.

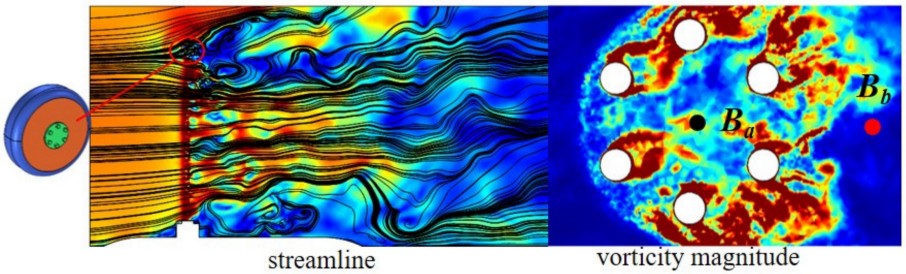

**Figure 15.** Flow characteristics of brake system.

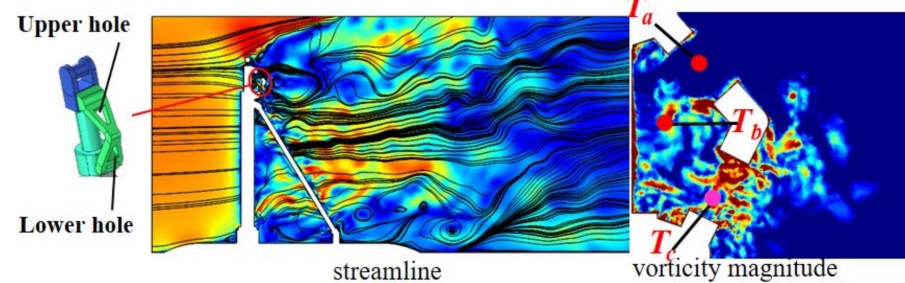

**Figure 16.** Flow characteristics of torque link.

### 4.2.1. Analysis of Flow Characteristics of Brake System

In order to further analyze the flow in these cavities, the center section of the brake system was selected to analyze the vorticity magnitude change at different time $t$. Figure 17 shows the distribution of vorticity magnitude at different wind speeds over time. Number the cylinders in the brake disc with 1–6.

From Figure 17, it can be concluded that the flow through the surface of the cylinder to produce boundary layer separation, and forms tail vortices behind the cylinder, and the tail vortices formed between the cylinders will interact. With the increase of speed, the vortex shedding frequency between the cylinders increases and the vortexes interaction increases, which makes the flow of the brake disc more complex. Figure 18a,b are graphs of pressure power spectral density at different wind speeds at the center point $B_a$ of the left and right brake system, respectively, where $f_a$, $f_b$, and $f_c$ are the frequencies corresponding to different wind speed. Figure 18c,d show the power spectral density of pressure fluctuation at point $B_b$ in the wake area of the brake system, respectively.

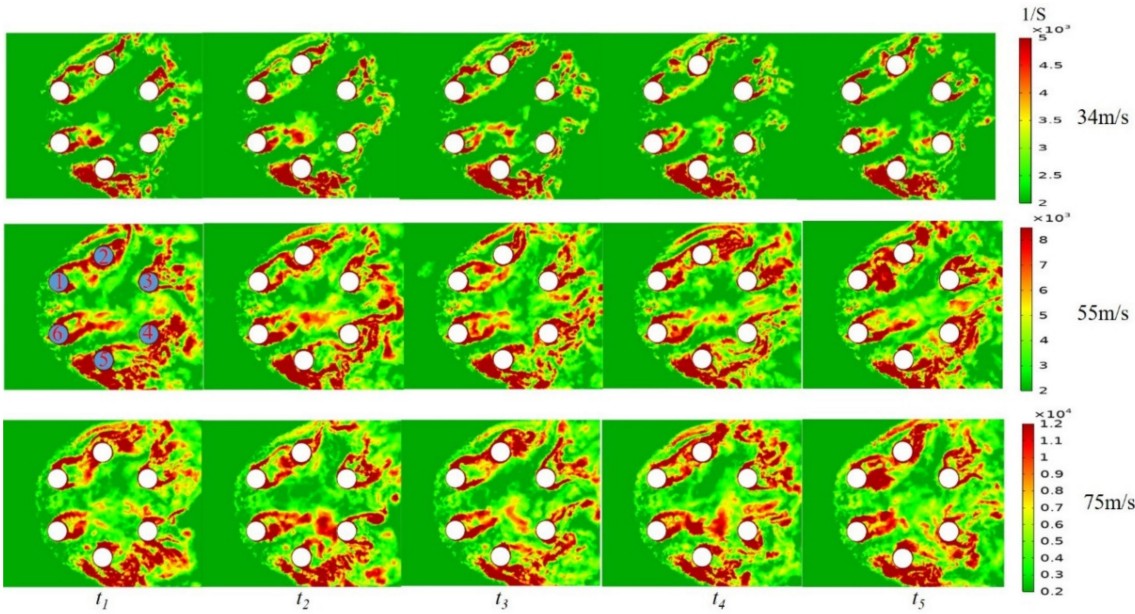

**Figure 17.** The vorticity magnitude distribution around the brake system.

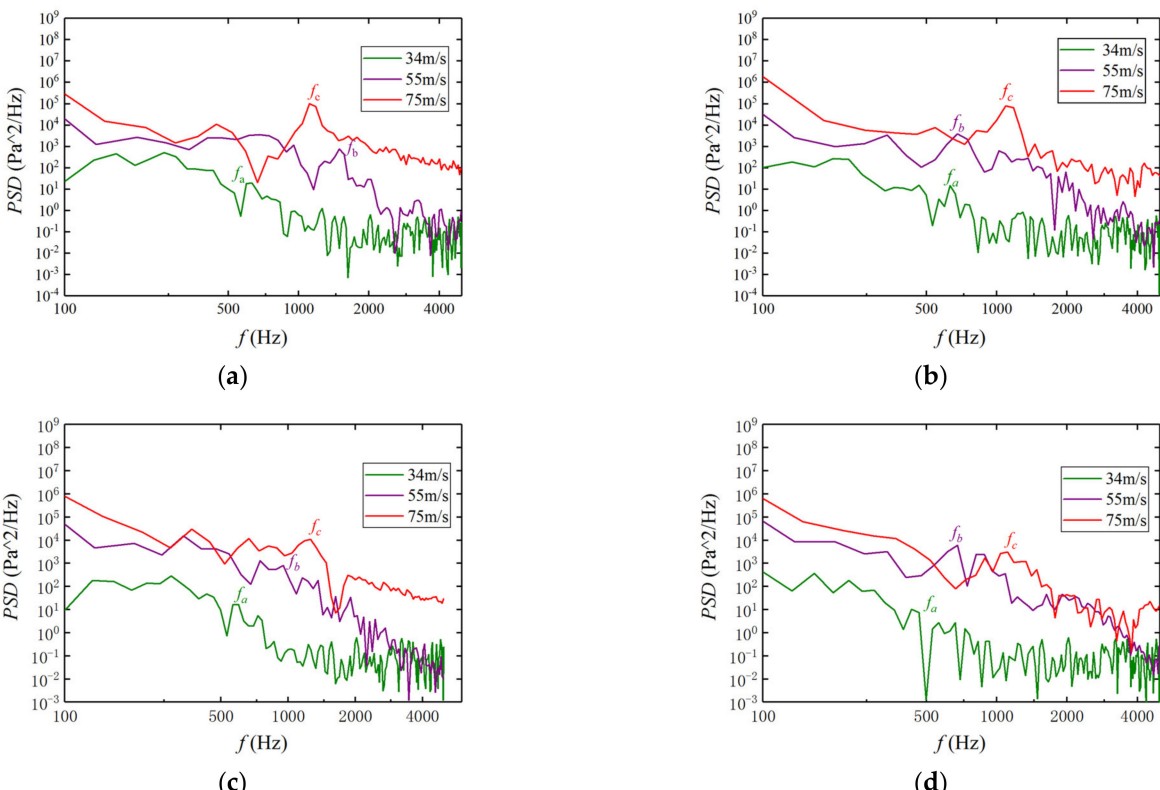

(**a**)

(**b**)

(**c**)

(**d**)

**Figure 18.** PSD diagram of pressure fluctuation at brake system. (**a**) At the point $B_a$ of the left brake system (**b**) At the point $B_a$ of the right brake system (**c**) At the point $B_b$ of the left brake system (**d**) At the point $B_b$ of the right brake system.

By analyzing Figure 18a,b it is concluded that when the speeds are 34 m·s$^{-1}$, 55 m·s$^{-1}$, and 75 m·s$^{-1}$, the frequencies corresponding to the amplitude of the pressure fluctuations at the center of the left and right brake system are $f_a$ = 631 Hz/636 Hz, $f_b$ = 680 Hz/1494 Hz, $f_c$ = 1091 Hz/1104 Hz. Except for 55 m·s$^{-1}$, the pressure fluctuation frequency of the left and right brake discs is roughly the same, and the frequency and amplitude of the fluctuations increase with the increase of speed. When the velocity reaches 75 m·s$^{-1}$, the interaction between the wake vortices of different cylinders increases, resulting in the

dissipation of vortex energy, so that the pressure amplitude increases slightly when the velocity is 55 m·s$^{-1}$~75 m·s$^{-1}$. From Figure 18c,d it can be found that in the wake area of left and right brake discs, when the speed increases from 55 m·s$^{-1}$ to 75 m·s$^{-1}$, the vortexes interaction are more obvious, resulting in the dissipation of vortex energy, so that the amplitude of pressure fluctuation basically does not increase at 75 m·s$^{-1}$.

### 4.2.2. Analysis of Flow Characteristics of Torque Link

Due to the existence of many cavities in the landing gear, the flow is complicated. Figures 19 and 20 are the flow and vorticity magnitude changes of the torque link at different wind speeds.

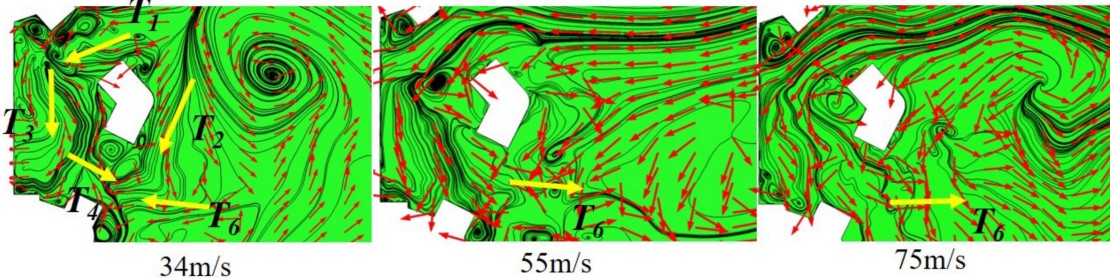

**Figure 19.** Streamline distribution at the cross section of the torque link.

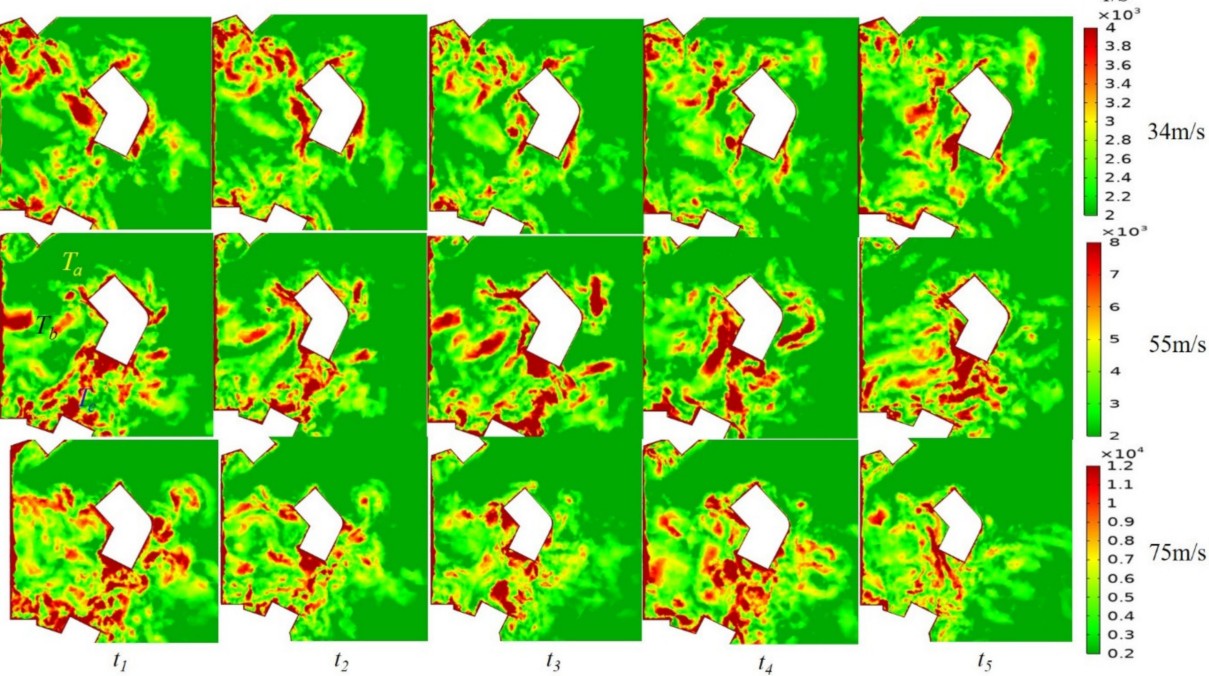

**Figure 20.** The vorticity magnitude changes with time at different speeds at the center section of the torque link.

According to Figure 19: I. After the fluid flows through the torque link, part of the fluid flows back to the cavity due to the pressure difference. The returned fluid is divided into two branches, $T_1$ and $T_2$. First, part of the fluid flows to $T_1 \rightarrow T_3 \rightarrow T_4$. Secondly, another part of fluid $T_2$ flows to the lower hole and interacts with $T_4$. II. In the upper hole area: as the speed increases, more air flows into the upper hole. In the lower hole area: (1) when the incoming flow velocity is 34 m·s$^{-1}$, the fluid $T_4$ cannot flow out smoothly due to the obstruction of $T_2$, resulting in large part of the fluid $T_6$ flows back to $T_4$. (2) When the speed is 55 m·s$^{-1}$~75 m·s$^{-1}$, $T_4$ is hindered by $T_2$ and gradually decreases as the speed increases, and most of the fluid in the cavity flows out along $T_4 \rightarrow T_6$.

From Figures 19 and 20, it can be concluded that: (1) When the speed is 34 m·s$^{-1}$, the incoming flow velocity is low, so the fluid flowing into the cavity is less, the vorticity magnitude is small, and the vorticity magnitude distribution is mainly concentrated in the upper region. When the speed increases to 55 m·s$^{-1}$ and 75 m·s$^{-1}$, the amount of fluid flowing into the cavity through the upper hole increases, and boundary layer separation vortices are formed on the wall surface. These vortices interact with each other in the lower region; (2) When the speed is 55 m·s$^{-1}$, the obstructive effect of $T_2$ on $T_4$ at the lower hole is still strong, making the fluid unable to be discharged smoothly and forming many vortices, resulting in no significant change in vorticity with the increase of time. (3) When the speed is 75 m·s$^{-1}$, because the obstructive effect of $T_2$ on $T_4$ is significantly reduced, a large amount of fluid flows out, so that the vortex in the lower hole area first increases and then decreases with time.

In order to further research the wind speed effect on the cavity flow characteristics of torque link, three points are selected as the upper hole point $T_a$, the middle point $T_b$, and the lower hole point $T_c$, and the pressure fluctuations are analyzed as shown in Figure 21. Table 3 shows the frequency corresponding to the peak value at different wind speeds at each point.

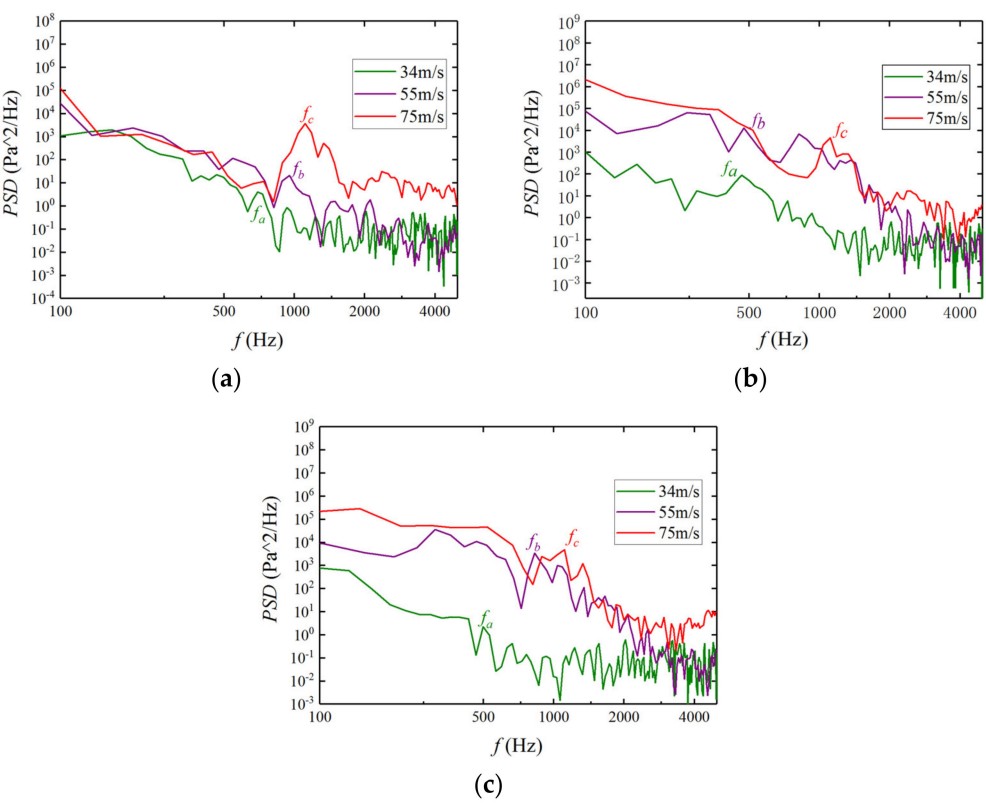

**Figure 21.** PSD diagram of pressure fluctuation at torque link. (**a**) At point $T_a$ (**b**) At point $T_b$ (**c**) At point $T_c$.

**Table 3.** Frequency corresponding to the peak value at different points.

|  | $f_a$/Hz (34 m·s$^{-1}$) | $f_b$/Hz (55 m·s$^{-1}$) | $f_c$/Hz (75 m·s$^{-1}$) |
|---|---|---|---|
| $T_a$ | 699 | 949 | 1109 |
| $T_b$ | 490 | 478/823 | 1109 |
| $T_c$ | 500 | 829 | 1113 |

According to Figure 21 and Table 3, it can be concluded that: (1) With the increase of speed at the three points, the frequency of pressure fluctuation increases, and the frequency of the three points is roughly the same at 75 m·s$^{-1}$; (2) At the point $T_a$, since the vortex is mainly concentrated in the upper half when the speed is 34 m·s$^{-1}$, the pressure fluctuates greatly, resulting in the same pressure fluctuation amplitude between 34 m·s$^{-1}$ and 55 m·s$^{-1}$. When the speed reaches 75 m·s$^{-1}$, a large amount of fluid flows in, and the pressure fluctuates obviously at $T_a$, so an obvious peak appears at 75 m·s$^{-1}$. (3) At the point $T_b$ and $T_c$, since the vorticity in this area is less at 34 m·s$^{-1}$, the pressure fluctuation amplitude is quite different from 55 m·s$^{-1}$ and 75 m·s$^{-1}$. From Figures 19 and 20, the obstructive effect of $T_2$ on $T_4$ is significantly reduced at the lower hole, and many vortex structures are discharged and dissipated at the lower hole. Therefore, the pressure fluctuation amplitude in the middle and lower regions at 75 m·s$^{-1}$ does not increase compared with that at 55 m·s$^{-1}$.

### 4.3. Far Field Noise Radiation

In the Section 4.2, the wind speed effect on the flow characteristics of cavities in the landing gear (brake system and torque link) are analyzed. In order to further study the wind speed effect on the aerodynamic noise generated by the complex flow of various components, the tire (A), brake system (B), torque link (C), and shock strut (D) were selected as the research objects (as shown in Figure 22).

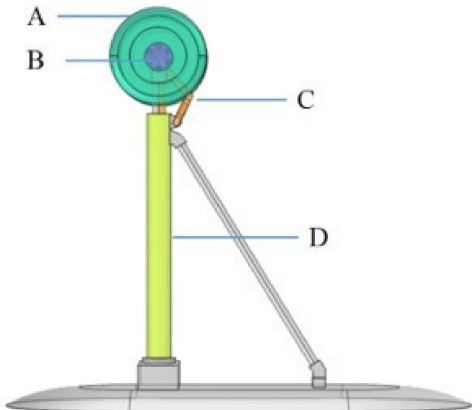

**Figure 22.** Parts of the landing gear.

The surface sound pressure of components A to D were extracted as sound sources, and calculate their aerodynamic noise characteristics at different wind speeds. A point 3 m away from the landing gear was selected to compare the SPL evolution with frequency of different parts under the same wind speed (as shown in Figure 23) and the SPL evolution with frequency of the same part under different wind speeds (as shown in Figure 24). Table 4 shows the OASPL of different components under different wind speeds.

From Figures 23 and 24 and Table 4, it can be concluded that: (I) Under different wind speeds, the contribution of A~D components' sound pressure level to the total noise is as follows: shock strut > tire > torque link > brake disc. (II) At the speed of 34 m·s$^{-1}$ to 55 m·s$^{-1}$, the contribution of each component to the total noise increases with the increase of the speed. In addition, the amplitude of the sound pressure level of the torque link and brake disc also increases significantly, which has a great impact on the total noise. (III) At the speed of 55 m·s$^{-1}$ to 75 m·s$^{-1}$, the amplitude of sound pressure level of torque link and brake disc increases slightly, and the contribution to the total noise decreases obviously. The increase of the total noise mainly comes from the main parts such as shock strut and tire.

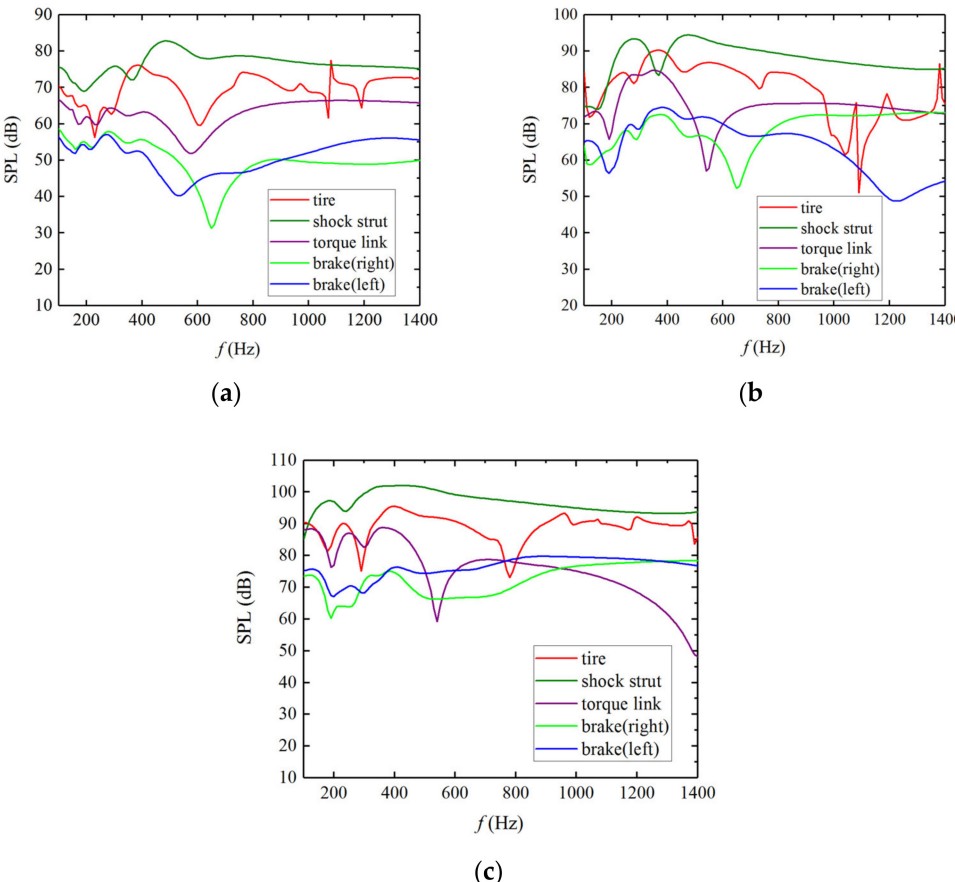

**Figure 23.** SPL evolution with frequency at different components. (**a**) 34 m·s$^{-1}$ (**b**) 55 m·s$^{-1}$ (**c**) 75 m·s$^{-1}$.

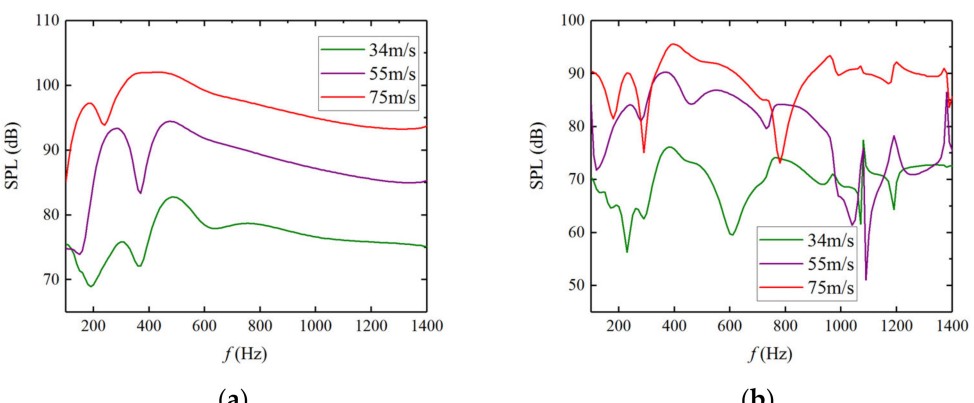

**Figure 24.** *Cont.*

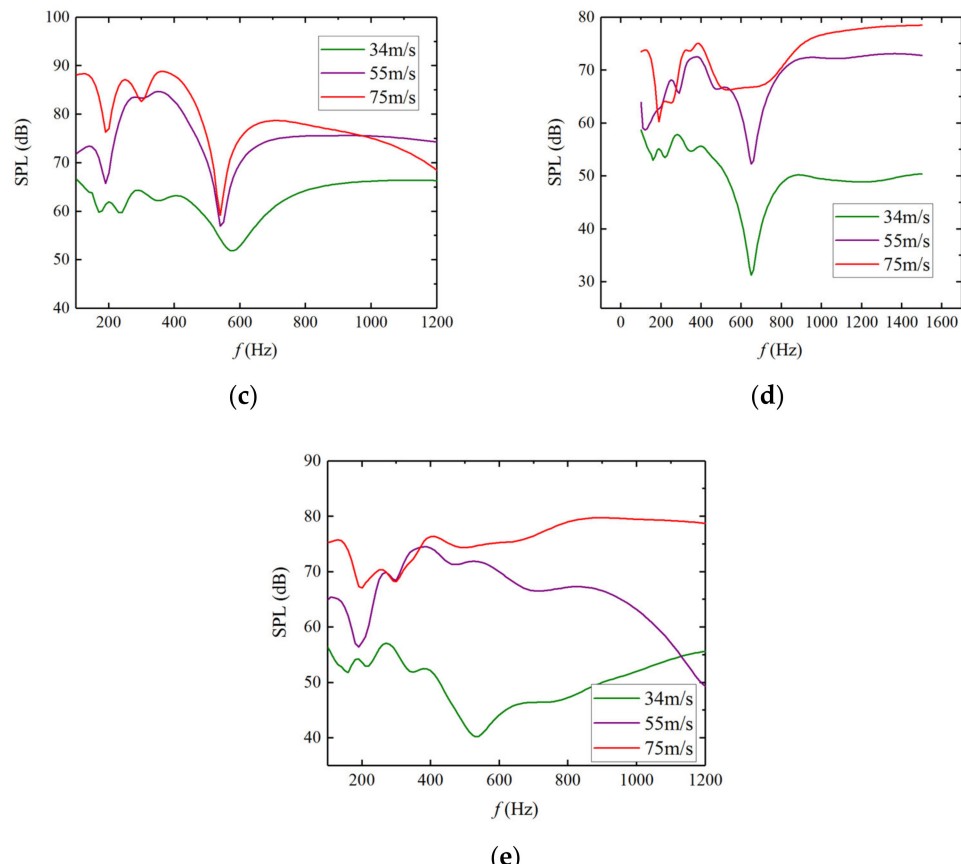

**Figure 24.** SPL evolution with frequency of the same part under different wind speeds. (**a**) Shock strut (**b**) Tire (**c**) torque link (**d**) brake (left) (**e**) brake (right).

**Table 4.** OASPL of each component at different wind speeds.

|            | Velocity               | Tire  | Shock Strut | Torque Link | Brake     |
|------------|------------------------|-------|-------------|-------------|-----------|
|            | 34 m·s$^{-1}$          | 92.7  | 98.8        | 85.8        | 74.5/73.9 |
| OASPL/dB   | 55 m·s$^{-1}$          | 104.6 | 110.9       | 98.6        | 90/91.1   |
|            | 75 m·s$^{-1}$          | 111.6 | 119.3       | 102.9       | 98.7/96.6 |

## 5. Conclusions and Future Work

Firstly, the numerical model of landing gear was established, and the feasibility of the simulation model was verified by experiments. Then according to the simulation results, the wind speed effect on the flow and acoustic characteristics of the minor cavity structures in a two-wheel landing gears were analyzed, and the following conclusions were drawn:

For the flow characteristics of the brake disc, at the wind speed of 55 m·s$^{-1}$~75 m·s$^{-1}$, the vortexes interaction is enhanced, resulting in vortex energy dissipation, so that the increment of pressure fluctuation amplitude is less than 34 m·s$^{-1}$~55 m·s$^{-1}$. For the flow characteristics of the torque link, with the increase of speed, the obstruction at the lower hole of the torque link decreases, and many vortices flow out of the lower hole and are dissipated, so that the pressure fluctuation amplitude of 75 m·s$^{-1}$ almost does not increase compared with 55 m·s$^{-1}$.

Furthermore, the sound pressure of each component was extracted, and their noise characteristics were analyzed under different wind speeds. It was found that: (I) The contribution of each component's sound pressure level under different wind speeds was as follows: shock strut > tire > torque link > brake disc. (II) At the speed of 34 m·s$^{-1}$ to

55 m·s$^{-1}$, the contribution of each component to the total noise increases with the increase of the speed. Moreover, the amplitude of the sound pressure level of the torque link and brake disc also increases significantly, which has a great impact on the total noise. (III) At the speed of 55 m·s$^{-1}$ to 75 m·s$^{-1}$, the increase of total sound pressure level mainly comes from the main components such as shock strut and tire, and the small components such as brake disc and torque link make very little contribution to the total noise.

In conclusion, this study investigated the effects of wind speed on the flow and acoustic characteristics in a two-wheel landing gear. In the next step, according to this research result, an optimal design scheme of landing gear noise reduction is expected to be proposed, with experiment to be conducted to do the validation. In addition, it is necessary to study other configurations, such as the four-wheel and the six-wheel, so as to achieve more understanding of the landing gear.

**Author Contributions:** L.H.; original paper writing and editing, numerical simulation and analysis. K.Z.; experimental testing, data analysis, project administration and financial support. J.L.; experimental test. V.K.; Experimental testing and writing instruction, I.B.; English writing instruction, T.Z.; Guidance on software use and parallel computing. All authors have read and agreed to the published version of the manuscript.

**Funding:** This work was sponsored by the National Natural Science Foundation of China, under the grant number 11902340.

**Institutional Review Board Statement:** Not applicable.

**Informed Consent Statement:** Not applicable.

**Data Availability Statement:** Not applicable.

**Acknowledgments:** The authors want to thank the Shujie Jiang and Chen He in CARDC for their help in numerical simulation.

**Conflicts of Interest:** The authors declare no conflict of interest.

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
