# Peer review of "A Numerical Study of the Wind Speed Effect on the Flow and Acoustic Characteristics of the Minor Cavity Structures in a Two-Wheel Landing Gear"

_applsci, doi:10.3390/app112311235_

Round 1

Reviewer 1 Report

The present paper can not be accepted, only a major revision that addresses all 41 remarks below would warrant a reconsideration:

1. I would reconsider the title as the acoustic content in the paper is rather "thin", it seems more inclined towards aerodynamics.

2. The first affiliation should be made complete, including the location, address, etc.

3. The contact email address in the first affiliation should be the official one, not a private one. If the authors can not provide the official one then he/she should explain the affiliation.

4. The second affiliation should be improved in the same way as for the first one. 

5. line 9: What is "aircraft body noise"? I think the normal way to refer to this is airframe noise.

6. The second sentence is strange, and it is useless to introduce acronyms as they are not used in the abstract again.

7. The introduction is rather short and does not properly introduce the topic and its context (see also few remarks below). Also, it does not properly indicate the novelty of the current work, the goal and does not provide a clear structure overview  (the latter two items are combined in the paper into one, but this is not good).

8. In the introduction, several important EU programs were not mentioned. For instance, VALIANT to name one. This must be improved.
Also, the authors need to properly mention that the is also much cylinder research that does not deal with circular cross-sections by square cross sections. Furthermore, many DES studies have been performed and should be mentioned (eg. on tandem configurations.)

9. Several important references wrt landing gear noise are not mentioned:

Dobrzynski et al, A European study on landing gear airframe noise sources, 6th Aeroacoustics Conference and Exhibit, 2000.

Dobrzynski et al, Research into landing gear airframe noise reduction, 8th AIAA/CEAS Aeroacoustics Conference & Exhibit, 2002.

Dobrzynski et al, Research at DLR towards airframe noise prediction and reduction, Aerospace Science and Technology, 2008.

Manoha et al, LAGOON: an experimental database for the validation of CFD/CAA methods for landing gear noise prediction, 14th AIAA/CEAS aeroacoustics Conference & Exhibit, 2008.

Dobrzynski et al, Experimental assessment of low noise landing gear component design, International Journal of Aeroacoustics, 2010.

10. Several important references wrt computational methods were not mentioned:

Ewert at al, The simulation of airframe noise applying Euler-perturbation and acoustic analogy approaches, International Journal of Aeroacoustics, 2005.

Bauer et al, Application of a discontinuous Galerkin method to discretize acoustic perturbation equations, AIAA journal, 2011.

Ewert et al, CAA broadband noise prediction for aeroacoustic design, Journal of sound and Vibration, 2011.

Greschner et al, Turbulence Modelling Effects on Tandem Cylinder Interaction Flow and Analysis of Installation Effects on Broadband Noise Using Chimera Technique, 30th AIAA Applied Aerodynamics Conference, 2012.

Knacke et al, Prediction of broadband noise from two square cylinders in tandem arrangement using a combined ddes/fwh approach, Turbulence and Interactions, 2014.

Dawi et al, Direct and integral noise computation of two square cylinders in tandem arrangement, Journal of Sound and Vibration, 2018.

Terracol et al, Numerical Wire Mesh Model for the Simulation of Noise-Reduction Devices, AIAA Journal, 2021.

11. In section 2: the coordinate system is not properly introduced or mentioned in the figures.

12. I find figures 1, 2, and 3 too small and not clear at all: this must be improved.

14. In section 3, line 111: Variables should be indeed in Italic, however, non.dimensional number should be in Roman. Also, the abbreviation of Mach number is M, not Ma. This is a common mistake, however, still wrong.

15. The "LES method"in section 3.1.1. is useless, as only the basics are stated and nowhere it is referred back to these. Also, although the basics are stated, several things are not properly introduced. For instance, what is the exact meaning of the overbar! This is rather important, especially if you are also talking about acoustics where you are only interested in the deviatory part of the pressure.

16. It does not become clear which code for the LES is used, is it in-house (reference to publication?) or a commercial code??

17. I think the mathematics is extremely sloppy, non-precise and in some moments simply wrong. For instance, line 138-145: what is the exact difference of rho_prime, rho_t and rho_0? What is the difference between rho_t and rho_prime? 

18. In Eq8 there are written 2 equations, so in line 142 it is "governing equationS"

19. The "momentum" equation has 3 components, this is wrong at current. This also applies to its "source term" Sm. So, the "governing equation" seems to be completely wrongly written there (I hope it is only an error in the text, not in the simulations itself?!?)

20. Line 154: the mesh distance of ".02 mm" is rather useless, give this in a wall unit dimension as the reader can then judge it.

21. Figure 5b is unclear and there is can not be seen at all what are the details.

21. In section 3.4: it is called solver, however, no solver is mentioned at all.

22. The information given in the section 3 is insufficient and as such the readers can not judge what is the quality of the results (or reproduce them). As such the results are to a large extend useless. This must be significantly improved!

23. The description in 4.1 is totally insufficient and at many places not to be understood. for instance: Line 184 "The inlet adopts velocity inlet, the outlet is set as pressure outlet, the cylindrical surface is non slip wall, the boundary parallel to the spanwise direction of the 
cylinder is set as no slip wall, and other boundaries are set as periodic flow boundary."

or line 186: " In the calculation domain, the grids around the cylinder and the wake area are encrypted, a..."

Nowhere is the span-wise extend given?!? 

24. In figure 7 nothing can be seen, either improve or remove this figure.

25. I find the discussion on figure 8 totally insufficient. Almost all critical issues are simply ignored, i.e., deviations between azimuthal angle between 80 and 280 degrees. 

26. Also in figure 8: degrees is not a unit so needs to be remove at the label of the x-axes.

27. lines 200-206: How the errors in percentage are obtained is puzzling. At theta =180 i see a difference of about 0.25 (0.5-0.25), which leads to a percentage error that is "slightly larger than the mere 4 to 5 percent you mention. Also the other errors seem to be completely wrong!  

28. The same applies to the complete "far field noise" section 4.1.2. It is rather vague and unclear what is displayed and simply presenting 3 lines with quite some deviation as a validation seem rather optimistic.

29. line 225: The frequency difference corresponding to the peak value of experiment and simulation are known to be quite insensitive and provide a much too positive image when using this as a validation criteria. SO, again: rather weak validation.

30. From here I will refrain from remarking everything as the authors need to improve the first part and also implement this in a similar way to the rest of the paper.

31. In figure 12: what velocity scaling comes out of this? Does it match literature.

32. Figures 14 and 15: must be improved, not clearly see are the structures and also the colour usage with the arrows is bad. Same applies to figures 18, 

33. Figure 16: Vorticity is a vector. Which component is shown or is it the magnitude? Also, the legend in the figure is too small to read. What are the units? This must be clear from the figure and also from the caption!. The same applies to figure 19.

34. Figure 20 is too small, the font can not be read. Same applies to figure 22. 

35. Figure 21 can be removed and combined with figure 1.

36. Section 5, which deals with the aeroacoustics, is only 20 lines long. I think this is rather small for a paper containing hundreds of lines and where aeroacoustics is prominently in the title. The authors need to change the title of the content of section 5.

37. I do not agree with the conclusion as should have become clear from the above remarks. For instance, the sentence "Firstly, the numerical model of landing gear is established, and the feasibility of the 396
simulation model is verified by experiments. " remains questionable.

38. On page 16: Several things appear twice on this page, for instance, two times "acknowledgement" (although two times with a slightly different title.

39. On page 16: The Acknowledgement should indicate the affiliation of the two persons who are specifically thanked.

40. At "Author Contributions":  I find the contribution of "BELYAEV Ivan " rather small (it is stated as "language editing service") which does not justify co-authorship. It is sufficient to only mention this person in the acknowledgement. Frankly, the contributions of the two persons now in the Acknowledgement (Dr. Shujie and Dr. Chen) seem much larger....

41. The format in the references list is not correct for many entries, eg. 3, 7, 16 and 22.

Author Response

Dear reviewer:

Thank you very much for the comment and remarks on this manuscript. Questions and suggestions you gave are very professional and important, which highly appreciate. In this letter, I will reply to all these questions, and the corresponding revisions have been also made in the revised manuscript.

I have made changes according to your opinions, as follows:

  1. I would reconsider the title as the acoustic content in the paper is rather "thin", it seems more inclined towards aerodynamics.

Thank you for your question. I agree with you very much and change the title to " A Numerical Study of the Wind Speed Effect on the Flow and Acoustic Characteristics of the Minor Cavity Structures in a Two-wheel Landing Gear"

  1. The first affiliation should be made complete, including the location, address, etc.

Thank you for your question. The author's information is supplemented

  1. The contact email address in the first affiliation should be the official one, not a private one. If the authors cannot provide the official one then he/she should explain the affiliation.

Thank you for your question. I am very sorry that my official email has not been applied for, so I only keep the email of the corresponding author

  1. The second affiliation should be improved in the same way as for the first one.

Thank you for your question. The information of second affiliation has been modified

  1. line 9: What is "aircraft body noise"? I think the normal way to refer to this is airframe noise.

Thank you for your question. Modify "aircraft body noise" to "airframe noise"

  1. The second sentence is strange, and it is useless to introduce acronyms as they are not used in the abstract again.

Thank you for your question. Acronyms have been deleted

  1. The introduction is rather short and does not properly introduce the topic and its context (see also few remarks below). Also, it does not properly indicate the novelty of the current work, the goal and does not provide a clear structure overview (the latter two items are combined in the paper into one, but this is not good).

Thank you very much for your question. We didn't describe the background of the landing gear in detail. By reading some literature, we rewritten the introduction, added important EU projects, and pointed out the innovation of this paper.

  1. In the introduction, several important EU programs were not mentioned. For instance, VALIANT to name one. This must be improved.

Also, the authors need to properly mention that the is also much cylinder research that does not deal with circular cross-sections by square cross sections. Furthermore, many DES studies have been performed and should be mentioned (eg. on tandem configurations.)

Thank you very much for your question. We didn't describe the background of the landing gear in detail. By reading some literature, we rewritten the introduction, added important EU projects, and pointed out the innovation of this paper.

  1. Several important references wrt landing gear noise are not mentioned:
  • Dobrzynski et al, A European study on landing gear airframe noise sources, 6th Aeroacoustics Conference and Exhibit, 2000.
  • Dobrzynski et al, Research into landing gear airframe noise reduction, 8th AIAA/CEAS Aeroacoustics Conference & Exhibit, 2002.
  • Dobrzynski et al, Research at DLR towards airframe noise prediction and reduction, Aerospace Science and Technology, 2008.
  • Manoha et al, LAGOON: an experimental database for the validation of CFD/CAA methods for landing gear noise prediction, 14th AIAA/CEAS aeroacoustics Conference & Exhibit, 2008.
  • Dobrzynski et al, Experimental assessment of low noise landing gear component design, International Journal of Aeroacoustics, 2010.

Thank you very much for listing the papers of many famous scholars in this field. I think as a scientific researcher in this field, we must read a lot of papers in this field, so we have added all the papers you put forward in this paper

  1. Several important references wrt computational methods were not mentioned:
  • Ewert at al, The simulation of airframe noise applying Euler-perturbation and acoustic analogy approaches, International Journal of Aeroacoustics, 2005.
  • Bauer et al, Application of a discontinuous Galerkin method to discretize acoustic perturbation equations, AIAA journal, 2011.
  • Ewert et al, CAA broadband noise prediction for aeroacoustic design, Journal of sound and Vibration, 2011.
  • Greschner et al, Turbulence Modelling Effects on Tandem Cylinder Interaction Flow and Analysis of Installation Effects on Broadband Noise Using Chimera Technique, 30th AIAA Applied Aerodynamics Conference, 2012.
  • Knacke et al, Prediction of broadband noise from two square cylinders in tandem arrangement using a combined ddes/fwh approach, Turbulence and Interactions, 2014.
  • Dawi et al, Direct and integral noise computation of two square cylinders in tandem arrangement, Journal of Sound and Vibration, 2018.
  • Terracol et al, Numerical Wire Mesh Model for the Simulation of Noise-Reduction Devices, AIAA Journal, 2021.

Thank you very much for listing the papers of many famous scholars in this field. I think as a scientific researcher in this field, we must read a lot of papers in this field, so we have added all the papers you put forward in this paper

  1. In section 2: the coordinate system is not properly introduced or mentioned in the figures.

Thank you for your question. The coordinate system is added in Figures 1 and 3. “O is the coordinate origin (the midpoint of the connecting line between the two tire centers of the landing gear), and U, V and W are the velocity components in X, Y and Z directions respectively.”

  1. I find figures 1, 2, and 3 too small and not clear at all: this must be improved.

Thank you for your question. Figure 1, figure 2 and figure 3 are improved

  1. In section 3, line 111: Variables should be indeed in Italic, however, non.dimensional number should be in Roman. Also, the abbreviation of Mach number is M, not Ma. This is a common mistake, however, still wrong.

Thank you for your question. Change the variable to Roman font and modify the abbreviation of Mach number

  1. The "LES method"in section 3.1.1. is useless, as only the basics are stated and nowhere it is referred back to these. Also, although the basics are stated, several things are not properly introduced. For instance, what is the exact meaning of the overbar! This is rather important, especially if you are also talking about acoustics where you are only interested in the deviatory part of the pressure.

Thank you for your question. Change the title of 3.1.1 to "governing equation". The upper horizontal bar represents the filtered average value. There are many mathematical errors in the equation, which have been modified by referring to the literature [25].

We just want to introduce the basic equation of LES, so we modify the title '3.2 LES method' to '3.2 governing equation'. There are indeed errors in the description of the governing equations, which have been modified in reference 25. LES model mainly adopts the LES module of COMSOL, Therefore, we added the software platform used.

Acoustic calculation is mainly based on the aeroacoustic module of COMSOL software, and LEE equation mainly refers to the software and corresponding literature [33,35]. We added references to the paper.

  1. Mason, P.J. Large-Eddy Simulation: A Critical Review of the Technique. Q.J Royal Met. Soc. 1994, 120, 1–26, doi:10.1002/qj.49712051503.
  2. Bauer, M.; Dierke, J.; Ewert, R. Application of a Discontinuous Galerkin Method to Discretize Acoustic Perturbation Equations. AIAA Journal 2011, 49, 898–908, doi:10.2514/1.J050333.
  3. Bissuel, A.; Allaire, G.; Daumas, L.; Barré, S.; Rey, F. Linearized Navier–Stokes Equations for Aeroacoustics Using Stabilized Finite Elements: Boundary Conditions and Industrial Application to Aft-Fan Noise Propagation. Computers & Fluids 2018, 166, 32–45, doi:10.1016/j.compfluid.2018.01.011.
  4. It does not become clear which code for the LES is used, is it in-house (reference to publication?) or a commercial code??

Thank you for your question. The numerical simulation is based on COMSOL software

We added an author, Tian Zhang; Guidance on COMSOL software use and parallel computing.

  1. I think the mathematics is extremely sloppy, non-precise and in some moments simply wrong. For instance, line 138-145: what is the exact difference of rho_prime, rho_t and rho_0? What is the difference between rho_t and rho_prime?

Thank you for your question. Where ρt, ut, and pt are the small perturbations exerted by density, velocity and pressure on the mean flow, respectively; ρ0, u0, p0 are the density, velocity and pressure of the flow field respectively.

Because the Linear Euler Equation linearizes the Euler equation, where the subscript 0 represents the variable of flow field, which mainly considers the influence of flow on sound propagation. If the flow is not considered, the flow field variable can be set to 0. The subscript t represents the acoustic variable

  1. In Eq8 there are written 2 equations, so in line 142 it is "governing equationS"

Thank you for your question. We numbered the two equations respectively. Like the continuity equation and momentum equation in the N-S equation, this equation are linearized N-S equation and belongs to the governing equation.

  1. The "momentum" equation has 3 components, this is wrong at current. This also applies to its "source term" Sm. So, the "governing equation" seems to be completely wrongly written there (I hope it is only an error in the text, not in the simulations itself?!?)

Thank you for your question. Change the variables in the equation into vector form

  1. Line 154: the mesh distance of ".02 mm" is rather useless, give this in a wall unit dimension as the reader can then judge it.

Thank you for your question. We use y+ to describe the first layer of grid, Y+< 10, 14, and 18 correspond to wind speeds of 34 m·s-1, 55 m·s-1 and 75 m·s-1 respectively

  1. Figure 5b is unclear and there is can not be seen at all what are the details.

Thank you for your question. Figure 5b has been improved

  1. In section 3.4: it is called solver, however, no solver is mentioned at all.

Thank you for your question. Change the title of Section 3.4 to "simulations parameters"

  1. The information given in the section 3 is insufficient and as such the readers can not judge what is the quality of the results (or reproduce them). As such the results are to a large extend useless. This must be significantly improved!

Thank you for your question. The third section has been modified according to your previous comments, and listed one by one from the mathematical model, establishment of calculation domain, meshing and boundary condition setting. I think this is the information provided by general numerical simulation.

  1. The description in 4.1 is totally insufficient and at many places not to be understood. for instance: Line 184 "The inlet adopts velocity inlet, the outlet is set as pressure outlet, the cylindrical surface is non slip wall, the boundary parallel to the spanwise direction of the

cylinder is set as no slip wall, and other boundaries are set as periodic flow boundary."

or line 186: " In the calculation domain, the grids around the cylinder and the wake area are encrypted, a..."

Nowhere is the span-wise extend given?!?

Thank you for your question. The coordinate system is added in Figure 7, and the corresponding description is modified as “the spanwise (the z direction in Figure 7) length is 4D,”

The description of boundary conditions is modified

“The inlet adopted velocity inlet(U= U, V=W=0), the outlet was set as atmospheric pressure (p=1 atm), the cylindrical surface was non slip wall (u=0), the boundaries perpendicular to the z direction were set as asymmetric boundary (usrc=udst), and other boundaries were set as slip wall(u·n=0).”

  1. In figure 7 nothing can be seen, either improve or remove this figure.

Thank you for your question. Figure 7 shows the calculation domain, boundary condition setting and overall grid division. It has been improved and the coordinate system has been added

  1. Also in figure 8: degrees is not a unit so needs to be remove at the label of the x-axes.

Thank you for your question. Figure 8 has been modified

  1. I find the discussion on figure 8 totally insufficient. Almost all critical issues are simply ignored, i.e., deviations between azimuthal angle between 80 and 280 degrees.

Thank you for your question. At 80 and 280 degrees of the downstream cylinder, there is a deviation between the simulation and the experiment, which is explained.

“It can be seen from the Fig.8 that the surface pressure coefficient of the up-stream cylinder is in good agreement with the QFF and BART test results, and the downstream cylinder surface pressure coefficient is closer to the QFF experimental results, This is consistent with the results of literature[40]. The main reason for the error at position 80° and 280°of the downstream cylinder is that the downstream cylinder is affected by the upstream cylinder, the flow is more complex, and the spanwise length in the simulation is inconsistent with the experiment.”

  1. lines 200-206: How the errors in percentage are obtained is puzzling. At theta =180 i see a difference of about 0.25 (0.5-0.25), which leads to a percentage error that is "slightly larger than the mere 4 to 5 percent you mention. Also the other errors seem to be completely wrong!

Thank you for your question. I would like to answer your question here:

I admit that I made a big mistake. I read the decimal point wrong in the process of calculating the percentage, resulting in such a wrong description. If the error between experiment and simulation is described quantitatively, I suddenly don't know how to describe it. So I referred to the paper you recommended:

LOCKARD D, KHORRAMI M, CHOUDHARI M, . 2007-3450 Tandem Cylinder Noise Predictions[J]. 2007.

  1. Lockard, “Summary of the Tandem Cylinder Solutions from the Benchmark problems for Airframe Noise Computations-I Workshop,” in 49th AIAA Aerospace Sciences Meeting including the New Horizons Forum and Aerospace Exposition, American Institute of Aeronautics and Astronautics. doi: 10.2514/6.2011-353.

This is a description of the comparison between simulation and experimental results, which I have referred to

My simulation results show that the pressure coefficient of the upstream cylinder is in good agreement with the QFF and BART test results, there are some differences at the inflection point of the downstream cylinder, and the experiments in other areas are in good agreement with the simulation. This is consistent with the results of Lockard scholars. We believe that the reason for the error is that the downstream cylinder is affected by the upstream cylinder, so the flow is more complex, and the spanwise length in the simulation is inconsistent with the experiment. This leads to a large deviation between the downstream cylinder and the experimental results.

The content of this paper is modified with reference to the literature:

“It can be seen from Fig.8 that the surface pressure coefficient of the upstream cylinder is in good agreement with the QFF and BART test results, and the downstream cylinder surface pressure coefficient is closer to the QFF experimental results. This is consistent with the results of literature[20]. The main reason for the error at position 80° and 280° of the downstream cylinder is that the downstream cylinder is affected by the upstream cylinder, so the flow is more complex, and the spanwise length in the simulation is inconsistent with the experiment.”

  1. The same applies to the complete "far field noise" section 4.1.2. It is rather vague and unclear what is displayed and simply presenting 3 lines with quite some deviation as a validation seem rather optimistic.

Thank you for your question. It is true that we did not explain the error of experiment and simulation, so we modified the paper as follows:

“The simulation results at the highest peak are in good agreement with the experimental results, but there are some differences in other positions. Furthermore, from Fig.8, it can be found that the downstream cylinder is affected by the upstream cylinder, and the numerical model is scaled in the spanwise direction relative to the experiments. This makes the pressure coefficient of the downstream cylinder different from the experimental results, resulting in differences between the simulation and experimental values in the far-field sound pressure level. Compared with the results of flow and acoustic, the numerical model is feasible.

  1. line 225: The frequency difference corresponding to the peak value of experiment and simulation are known to be quite insensitive and provide a much too positive image when using this as a validation criteria. SO, again: rather weak validation.

Thank you very much for your question. By comparing the pressure coefficient and far-field noise of tandem cylinders, there is a large error in the pressure coefficient of downstream cylinders, which is consistent with the results of D. Lockard; In terms of far-field noise, the amplitude and frequency at the highest peak are in good agreement with the simulation, and there are some deviations at other frequencies. We explain the existing deviations.

“The simulation results at the highest peak are in good agreement with the experimental results, but there are some differences in other positions. Furthermore, from Fig.8, it can be found that the downstream cylinder is affected by the upstream cylinder, and the numerical model is scaled in the spanwise direction relative to the experiments. This makes the pressure coefficient of the downstream cylinder different from the experimental results, resulting in differences between the simulation and experimental values in the far-field sound pressure level. Compared with the results of flow and acoustic, the numerical model is feasible.”

For the verification of landing gear, we would like to further explain to you as follows:

I added a microphone comparison result, and referred to the simulation results in other papers, So I would like to explain my views: (1) the landing gear model does not make any simplification and looks relatively complex. I want to study the model which is not scaled, so the model is relatively large, and the number of grids may not be satisfied, but the computing resources are very limited. (2) Sound is generated by pressure pulsation. Large eddy simulation can analyze small vortex structure, which makes it very complex to solve sound source information in large eddy simulation, and mixed sound is easy to appear. (3) Because my simulation experience is not rich and my scientific research ability needs to be improved, I think it is still difficult to calculate the aerodynamic noise in numerical terms. From the results of flow around double cylinders, although there are some errors, it can still describe the variation law of flow field and noise. (4) I agree with you very much on the comparison between simulation and experiment, but the results of series cylinders basically show that this method is feasible. From the comparison results of landing gear, it is difficult to obtain the consistency of frequency analysis, therefore, this paper does not describe aerodynamic noise too much, and only discusses the variation law of landing gear noise amplitude at different wind speeds.

Here are the results of some reference articles:

[1]   S. Redonnet, G. Cunha, and S. Ben Khelil, “Numerical Simulation of Landing Gear Noise via Weakly Coupled CFD-CAA Calculations,” presented at the 19th AIAA/CEAS Aeroacoustics Conference, Berlin, Germany, May 2013. doi: 10.2514/6.2013-2068.

Giret, J.-C.; Sengissen, A.; Moreau, S.; Jouhaud, J.-C. Prediction of LAGOON Landing-Gear Noise Using an Unstructured LES Solver. In Proceedings of the 19th AIAA/CEAS Aeroacoustics Conference; American Institute of Aeronautics and Astronautics: Berlin, Germany, May 27 2013.

Redonnet, S.; Ben Khelil, S.; Bulté, J.; Cunha, G. Numerical Characterization of Landing Gear Aeroacoustics Using Advanced Simulation and Analysis Techniques. Journal of Sound and Vibration 2017, 403, 214–233, doi:10.1016/j.jsv.2017.05.012.

Ribeiro, A.F.; Casalino, D.; Fares, E.; Noelting, S.E. CFD/CAA Analysis of the LAGOON Landing Gear Configuration. In Proceedings of the 19th AIAA/CEAS Aeroacoustics Conference; American Institute of Aeronautics and Astronautics: Berlin, Germany, May 27 2013.

  1. From here I will refrain from remarking everything as the authors need to improve the first part and also implement this in a similar way to the rest of the paper.

Thank you for your question. I hope I can get your approval through my modification of the paper

  1. In figure 12: what velocity scaling comes out of this? Does it match literature.

Thank you for your question. I seldom see comparative analysis in this aspect and draw corresponding conclusions, but our experimental equipment is very advanced and the reliability of the results is verified. I think it is credible to draw such a result

  1. Figures 14 and 15: must be improved, not clearly see are the structures and also the colour usage with the arrows is bad. Same applies to figures 18,

Thank you for your question. Figure 14, Figure 15 and Figure 18 are improved

  1. Figure 16: Vorticity is a vector. Which component is shown or is it the magnitude? Also, the legend in the figure is too small to read. What are the units? This must be clear from the figure and also from the caption!. The same applies to figure 19.

Thank you for your question. Figures 16 and 19 should be " vorticity magnitude ", with the unit of 1/S. Figure 16 and figure 19 are improved

  1. Figure 20 is too small, the font can not be read. Same applies to figure 22.

Thank you for your question. Figure 20 and Figure 22 have been improved

  1. Figure 21 can be removed and combined with figure 1.

Thank you for your question. Your suggestion is very good. I think it can be retained because the content is too far away from Figure 2 for readers to read

  1. Section 5, which deals with the aeroacoustics, is only 20 lines long. I think this is rather small for a paper containing hundreds of lines and where aeroacoustics is prominently in the title. The authors need to change the title of the content of section 5.

Thank you for your question. I quite agree with you and change the title of Section 5 to "far field noise radiation",and section 5 is incorporated into Section 4.3

  1. I do not agree with the conclusion as should have become clear from the above remarks. For instance, the sentence "Firstly, the numerical model of landing gear is established, and the feasibility of the 396

simulation model is verified by experiments. " remains questionable.

Thank you for your question. There are differences in the experiment and simulation. We explain the errors. Because the experimental and simulation frequencies of the landing gear are not in good agreement, but there is little difference between simulation and experiment in amplitude. Therefore, this paper mainly studies the amplitude change law of wind speed on pressure and far-field sound radiation.

  1. On page 16: Several things appear twice on this page, for instance, two times "acknowledgement" (although two times with a slightly different title.

Thank you for your question. One of the acknowledges was deleted

  1. On page 16: The Acknowledgement should indicate the affiliation of the two persons who are specifically thanked.

Thank you for your question. We added their identity information

“The authors want to thank the editor and anonymous reviewers for their helpful comments and suggestions, and the Dr. Shujie Jiang (CARDC) and Dr. Chen He (CARDC) for their help in nu-merical simulation.”

  1. At "Author Contributions": I find the contribution of "BELYAEV Ivan " rather small (it is stated as "language editing service") which does not justify co-authorship. It is sufficient to only mention this person in the acknowledgement. Frankly, the contributions of the two persons now in the Acknowledgement (Dr. Shujie and Dr. Chen) seem much larger....

Thank you for your question. Maybe I made a mistake. Belyaev Ivan was a participant in the project and provided guidance in thesis writing and language editing. Modify the contents of the paper

  1. The format in the references list is not correct for many entries, eg. 3, 7, 16 and 22.

Thank you for your question. Malformed references have been modified

Sincerely,

Longlong Huang

Reviewer 2 Report

This study presents the experimental and FE simulation results for aerodynamic noise characteristics according to the change of wind speed for the landing gear, and these results can be usefully used for engineering design. However, academic originality does not seem to be high in terms of general test methods and presentation of finite element results. 1. It seems necessary to explain what other types of landing gear there are other than the landing gear presented in this study. 2. It is necessary to suggest what academic originality and improvements are produced, comparing with existing studies in the experimental methods (BART, QFF, etc.) and finite element analysis presented in this study. 3. In Figure 9, there are various peaks in numerical simulation that are not seen in Exp (QFF), which needs explanation. 4. In Figure 10, 14, 15, etc., it is necessary to add information about axis, legend and units. 5. After section 4.3, only finite element analysis results are presented, so it is difficult to know how this result differs from the experimental results. 6. In Figure 17, fa, fb, and fc seem difficult to confirm, please add more explanation. 7. With respect to Figures 22 and 23, various peaks are formed for each frequency band, and an explanation for this needs to be added. 8. In conclusion, it is necessary to explain in relation to the actual landing gear noise and damage type.

Reviewer 3 Report

Dear authors,
Please find my comments about you article hereunder.

Overall
Improve titles of tables and figures to clearly explain what is displayed.
line 423, Acknowledgements.
Units should not be placed after a symbol "/".

Abstract
Check the punctuation.

1 Introduction
Define abbreviations DES, LES, LBM, FW-H.
Briefly explain why you specifically use LES + LEE. Is the "best" model?
Briefly explain what kind of geometry you are going to study? Is it a new one? Or do you target a specific phenomenon such as the interactions of the vortices? Where is the novelty?
You briefly introduce all the sections except sections 2 and 6.

2 Wind tunnel test of landing gear
Change the title to specify that you present the setup of the test, and not the results.

3 Numerical simulations of landing gear
Change the title to specify that you present the setup of the simulation, and not the results.
Change title of 3.1 to calculation method of aerodynamics (to match 3.2).
Change title of 3.1.1 Build flow model, it does not mean anything (do you mean, description of the simulation domain?).
3.1.1 LES model, section numbering duplicated.
Line 110, D is undefined, do you mean D_L? if so define it directly.
Line 113, since the velocity varies over a wide range, give both the lower and upper bounds for Re number.
Change title of fig 4 to flow domain.
Line 118, briefly explain how you obtain the LES equations, or give a reference.
Line 120, there is an issue with the symbol "rho".
Line 128, issue with "nu_sgs", should it be bold or not?
Section 3.2, why do you describe Lighthill model here, since you do not use it. Move it to the intro, if relevant, and add a reference.
Section 3.2, as for the LES, briefly explain how you obtain the LEE, or give a reference.
Fig 5b, clarify which cross-sections are being visualized.
Section 3.3, add a detailed view of the prismatic mesh in the boundary layer.
Section 3.4, what are a "velocity condition" and an "open condition"? Clarify which variables/gradients are imposed for the flow.
Section 3.4, give references for perfect matching layer and hard sound field boundary (these are very specific).
The first sentence of the last paragraph (line 168-169) is not clear. How can you use the solution at 0.3s to initialize the solution at 0.1s.
Line 170, how did you determine the time step.

4 Results analysis
I would NOT put section 4.1 in chapter 4. The reason is that you already presented the wind tunnel and numerical setup for the LG (chapter 2 and 3), and section 4.1 talks about something else (other model and setup).
Figure 11, 12, 17, 20, 22, 23, units should not be put after a "/".
line 310, subsection 2), correct numbering.
Sentence line 303-304 is unclear.
line 339, "in order to further the flow", what do you mean?

Sections 5 and 6
The paper is mainly dedicated to analyzing the flow around the landing gear so that noise production can further be identified, analyzed and eventually lowered. However, few results are given on noise production, and no insights are given. Indicate what insights your results about noise production give, and how they can be used for further research, or LG design.

Best Regards,

Reviewer 4 Report

The paper focuses on the noise generated by the landing gear structure during aircraft operations. The introduction considers some research in the area, although it could be further improved. Methods are well described and having both numerical and experimental results is a huge plus. Furthermore, conclusions are supported by the results. On a final note, some formatting aspects, such as spacing between author’s names and references, tables caption location, its referencing and phrase punctuation should be addressed.

The paper should be accepted after a minor revision.

Round 2

Reviewer 1 Report

I think the authors have adapted most of my remarks, however, I noticed several errors in the references that have been added (and which must be corrected):

-In your revised document, the Reference 3 and Reference 6 are to my opinion the same, just in Reference 3 the author list is missing. This must be corrected. 

-Also, several of the references I have strongly suggested have not been implemented or added. It is rather misleading of the authors to mention in the reply that all has been added, however, several have not been added. This is rather bad. For completeness, I repeat here the ones that are missing:

Ewert at al, The simulation of airframe noise applying Euler-perturbation and acoustic analogy approaches, International Journal of Aeroacoustics, 2005

Ewert et al, CAA broadband noise prediction for aeroacoustic design, Journal of sound and Vibration, 2011.

Greschner et al, Turbulence Modelling Effects on Tandem Cylinder Interaction Flow and Analysis of Installation Effects on Broadband Noise Using Chimera Technique, 30th AIAA Applied Aerodynamics Conference, 2012.

Dawi et al, Direct and integral noise computation of two square cylinders in tandem arrangement, Journal of Sound and Vibration, 2018.

I think all 4 references above must be properly added, either in the introduction or at places where the validation is discussed.

-My previous remark 31: "In figure 12: what velocity scaling comes out of this? Does it match literature." has not been answered properly. Can you provide the velocity scaling based on your data?

Reviewer 3 Report

Dear authors,

Thank you for taking my comments into account and providing a revised version of the manuscript. I have a few minor, but important, comments:

  • Line 96, undefined reference.
  • Please define all the abbreviations, even if the are shown in the Figures. For example: on Fig9, what is PSD (power spectral density)? on Fig11, what is SPL?
    These should be mentioned and defined in the text. For example, "Fig9 shows the evolution of the PSD (power spectral density) with...".
  • Please check one last time the caption of all the tables and figures (not the title of the manuscript). For example Fig11 could be captioned: "Comparison of the evolution of the SPL with the frequency between the test data the results of simulation 10."
  • Please double check the symbols "/". Figure 11a still contains a "/" and it should not.
  • Line 268, "LES can capture small flow structures".
  • Conclusion, the conclusion should include a "next steps" paragraph where you explain how the results you obtain can or will be used in subsequent studies.
  • Acknowledgments, my previous comments mainly referred to a spelling mistake. You do not need to include the editor and reviewers within that section.

I would kindly ask you to take these comments into account in order to further improve your manuscript before final submission.

Best Regards,
